

# Spatial variability of mean daily estimates of actual evaporation from remotely sensed imagery and surface reference data

Robert N. Armstrong[1], John W. Pomeroy[2], Lawrence W. Martz[2]

[1]Queensland Alliance for Agriculture and Food Innovation, The University of Queensland, Toowoomba, 4350, Australia

[2]Centre for Hydrology, University of Saskatchewan, Saskatoon, S7N 1K2, Canada

*Correspondence to*: R. Armstrong (r.armstrong1@uq.edu.au)

**Abstract.** Land surface evaporation has considerable spatial variability that is not captured by point scale estimates

calculated from meteorological data alone. Knowing how evaporation varies spatially remains an important issue for improving parameterisations of land surface schemes and hydrological models, and various land management practices. Satellite-based and aerial remote sensing has been crucial for capturing moderate to larger scale surface variables to indirectly estimate evaporative fluxes. However, more recent advances for field research via unmanned aerial vehicles (UAVs) now allows for the acquisition of more highly detailed surface data.

Integrating models that can estimate *actual evaporation* from higher resolution imagery and surface reference data would be valuable to better examine potential impacts of local variations in evaporation on upscaled estimates. This study introduces a novel approach for computing a *normalised* index from surface variables that can be used to obtain more realistic distributed estimates of actual evaporation. For demonstration purposes the Granger and Gray evaporation model (G-D) was applied at a complex parkland site in central Saskatchewan, Canada. Visible

and thermal images and meteorological reference data required to parameterise the model was obtained at midday.

Normalised indexes (simple ratios) were computed at midday for albedo and net radiation. This allowed for single measured values albedo and mean daily net radiation to be scaled across high resolution images over a large study region. Albedo and net radiation estimates were within 5 – 10% of measured values. An evaporation estimate for a grassed surface was 0.5 mm larger than eddy covariance measurements. The methods applied have two key

advantages for estimating evaporation over previous remote sensing approaches, 1. Detailed daily estimates of *actual evaporation* were directly obtained using a physically-based evaporation model, and 2. Analysis of more detailed and reliable evaporation estimates may lead to improved methods for upscaling evaporative fluxes to larger scales.

## 1  Background and introduction

'Actual' evaporation is the water vapour physically transferred from a surface (e.g. plants, soil or water) to the atmosphere over a given time period (e.g. hourly or daily). Reliable estimates of actual evaporation are often needed over large spatial scales for applications such as water resource management, agriculture, ecology, and forecast modelling of weather and climate. However, estimates (or measurements) are often calculated at point



scales with footprints that can range from centimetres to several kilometres or more (Brutsaert, 1982). Consequently, point scale footprints in heterogeneous landscapes may contain large variability that needs to be considered more appropriately.

From an ecological standpoint, heterogeneous landscapes are comprised of distinct topographic features, land cover types, biological attributes and other physical properties that exhibit observable patterns in the order of meters (e.g. Yates et al., 2006; Zhang and Guo, 2007). Therefore, variable surface properties and state conditions exert a strong control on local surface energy fluxes. As a result, "scaling" evaporation estimates over large areas must consider a potential loss of information due to upscaling processes. The potential impacts of spatial variability on larger scale estimates of evaporation is still not well understood, and previous estimation and scaling methods have not examined this issue in detail.

For example, hydrologic and atmospheric modelling applications often require large to regional scale evaporation estimates but the underlying variability is difficult to examine practically (e.g. Avvisar and Pielke (1989), Baldocchi (2005), Brutsaert (1998), Claussen (1991; 1995), Klaassen (1992), Klaassen and Claussen (1995). Courault et al. (2005) and Gowda et al. (2007) have also reviewed remote sensing approaches which integrate surface images to derive key variables need to parameterise various energy-balance type evaporation models. This generally results in estimates at moderate scales of input images (e.g. Bisht et al., 2005). Purely empirical methods have also correlated evaporation with vegetation indices (e.g. Nagler et al., 2005).

A common remote sensing method has been to calculate evaporation indirectly as a residual of a simplified energy balance (e.g. Jackson et al., 1977; Seguin et al., 1989; Bussières et al., 1996). In such cases surface temperatures derived from thermal imagery is a critical input. More complex resistance-type formulations also exist based on developments by Monteith (1965); e.g. Norman et al. (1995), Anderson et al. (1997), Boegh et al. (2002), Houborg and Soegaard (2004), and Anderson et al. (2007). However, such approaches are computationally intensive and parameterising the resistance terms is difficult without detailed data.

Colaizzi et al. (2006) reviewed scaling approaches based on an evaporative fraction determined through complex solar radiation modelling. Mu et al., (2007) and Fisher et al. (2008) discuss advanced methods for deriving global scale estimates based on expert knowledge and detailed data sets obtained from the Ameriflux network in association with a global Fluxnet (Baldocchi et al., 2002). As an alternative to more complex methods, Granger (2000) integrated a complementary feedback approach with Penman's combination model and remote sensing imagery that can directly estimate actual evaporation, even in data sparse regions.

Previous methods have been valuable for integrating generalised representations of moderate to larger scale variability. However, most methods fail to include the fundamental interactions governing the evaporation process. More importantly, few (if any) have addressed the issue of how surface variability may impact upscaled evaporation estimates over large areas. The goal of this study is to investigate spatial variations in key surface variables driving evaporation and their general impact on upscaled estimates.

This work included two objectives, 1.) Use the spatial variability captured from one-time-of-day visible and thermal images at midday to distribute energy-balance and aerodynamic terms that drive combination evaporation models, and 2.) Examine impacts of spatial variations of underlying surface variables on daily estimates of evaporation, including smaller point-scale and larger areal estimates.



Methods applied here integrated high resolution aerial imagery collected on Aug 5, 2007 over a complex prairie landscape and surface reference data measured at a point, a physically-based evaporation model, and GIS. Estimates of mean daily 'actual' evaporation were calculated with a complementary feedback model introduced by Granger and Gray (1989). The model is well suited for a variety of Canadian environments (e.g. agricultural, prairie and boreal) and does not generally require detailed soil moisture information (except under severe moisture stress).

The methods are expected to be applicable to images taken from cameras and sensors aboard satellites, planes and unmanned aerial vehicles (UAVs). Due to technological advances there has recently been a tremendous increase in application of UAVs for field research, particularly in regards to agricultural crop improvements through field-based phenotyping (e.g. Chapman et al., 2014; Yang et al., 2017)

## 2    Study area

A case study was conducted on Aug 5, 2007 at St. Denis National Wildlife Area (SDNWA) in the parkland region of central Saskatchewan (see Armstrong et al., 2008). The landscape was characterised by hummocky, gently rolling terrain, and a few slopes of up to 10 – 15%. Elevations range from 540 m to 565 m and land use consisted of mixed cool season grasses, brome grass, cultivated land, and wetlands surrounded by tree rings or dense grasses. The soil region was classified as dark brown chernozem and soil texture is predominately silty loam (van der Kamp et al., 2003).

## 3    Data and methods

### 3.1    Granger and Gray evaporation model (G-D model)

Granger and Gray (1989) developed the G-D model from the complementary relationship of Bouchet (1963) and Penman's (1948) combination model. The G-D model extends the potential evaporation model to non-saturated surfaces using the relative evaporation term, $G$; defined as a ratio of actual to potential evaporation. The theory behind this is that as a surface dries the water availability is reduced but 'potential' evaporation increases due to a subsequent rise in surface temperature.

Integration of $G$ eliminated the need for observations of surface temperature and vapour pressure. As a result, estimates of actual evaporation were obtainable for non-saturated surfaces with atmospheric data alone (Granger, 1989). The equation can be stated as:

$$E = \frac{\Delta G (Q^* - Q_g) + \gamma G E_A}{\Delta G + \gamma} \ . \tag{1}$$

The available energy term is driven by net radiation, $Q^*$ (W m$^{-2}$) calculated as a sum of the net shortwave and longwave radiation components, the ground heat flux, $Qg$ is (W m$^{-2}$) which is assumed to be negligible (balances) for daily estimates, and slope of the saturation vapour pressure curve, $\Delta$.

The aerodynamic term includes the psychrometric constant, $\gamma$, and "drying power of the air", $E_A$, calculated using a Dalton type formula:





$$E_A = f(u)(e_a^* - e_a) \ ,$$
(2)

where $f(u)$ is a wind function, and the atmospheric vapour pressure deficit at 2 m height is derived from $e^*_a$ (saturated) and $e_a$ (actual). Pomeroy et al. (1997) empirically derived $f(u)$ as a function of wind speed and aerodynamic roughness height, $z_0$ (m) from extensive field data collected for prairie, boreal forest, and northern cold region environments in western Canada:

$$f(u) = 8.19 + 22Z_0 + (1.16 + 8Z_0)u \ ,$$
(3)

where $u$ is the mean daily wind speed (m s$^{-1}$).

The G-D complementary feedback method is driven by the non-linear relationship between $G$ and the relative drying power of the air, $D$:

$$G = \frac{1}{0.793 + 0.2e^{4.902D}} + 0.006D \ ,$$
(4)

where $D$ is a function of the humidity deficit and available energy stated as:

$$D = \frac{E_A}{E_A + \frac{(Q^* - Q_g)}{\lambda}} \ ,$$
(5)

and $\lambda$ is the latent heat of vaporisation (kJ kg$^{-1}$).

Armstrong et al. (2008) evaluated three physically-based point scale evaporation models over mixed grasses at the study area under conditions of non-limiting soil moisture. Estimates obtained with the G-D model compared well with eddy covariance measurements and was less data intensive than the Penman-Monteith model which also performed well. Armstrong et al. (2010), however, found the use of appropriate soil moisture constraints was required to obtain reliable estimates with the G-D model under drought conditions.

### 3.2 Field observations

Inputs needed to parameterise Eq. (1-5) included mean daily net radiation, air temperature, humidity, wind speed, and surface roughness heights. Outgoing radiation components were derived from high resolution digital and thermal images taken on Aug 5, 2007 during a Cessna flight at midday. Incoming radiation and atmospheric reference data were measured at two locations which could be used as either reference or validation sites depending on local weather conditions. Net radiation observations at an additional location provided a second site for validating estimated values.

Figure 1 shows the station locations, including one externally operated by the National Water Research Institute (Environment Canada, Saskatoon). Air temperature, humidity, wind speed, incoming radiation and surface temperature were recorded as 15 min averages. Incoming and outgoing shortwave and longwave radiation were measured with a CNR1 net radiometer (Kipp and Zonen, Delft, The Netherlands). Air temperature and humidity were measured at 2 m height using a Vaisala HMP45C (Campbell Scientific, Inc. Logan, Utah). An Exergen infrared temperature sensor, IRTC (Exergen, Watertown, Massachusetts) was used to measure surface temperature.

Canopy spectral reflectance was independently sampled on Aug 21, 2007 for validating albedo estimates derived from the visible images taken on Aug 5. Canopy reflectance was collected according to the methods of



Disney et al. (2004) and Zhang and Guo (2007). Samples were taken at 4.5 m intervals along a site transect (see Fig. 1) at 1 m height at nadir (25° field of view) with an ASD FR Pro spectroradiometer (Analytical Spectral Devices, Inc. Boulder, Colorado); spectral range of 350–2500 nm with 1 nm resolution. Samples were taken between 12 noon and solar noon local time. Reflectance was recalibrated every 10 min using a white spectralon
reflectance panel (Labsphere Inc. North Sutton, New Hampshire).

Eddy covariance (EC) observations were taken at approximately 2 m height with a three-dimensional sonic anemometer, CSAT3 (Campbell Scientific, Inc. Logan, Utah) and an ultraviolet krypton hygrometer, KH20 (Campbell Scientific, Inc. Logan, Utah). The EC data was corrected using a planar-fit axis rotation and correction algorithm (Wilczak et al., 2001).

### 3.3    Deriving key surface variables from one-time-of-day images

#### 3.3.1    Theoretical basis for a normalised index of relative ratios

Remotely sensed images contain valuable information that characterise highly variable surface properties (e.g. reflectance, temperature, RGB and greyscale DNs, etc.). Theoretically, relative variations in these properties can be quantified using a "normalised" index of relative ratios. For example, here we consider the 'evaporation ratio'
($E_R$) defined simply as the ratios of individual evaporation rates at different spatial locations ($E_i$) to a reference rate obtained at a specified location ($E_{ref}$):

$$E_R = \frac{E_i}{E_{ref}} \ . \tag{6}$$

Therefore, at the reference location and any other locations where $E_i = E_{ref}$, $E_R = 1$ but would vary from unity at all other spatial locations.

For obvious reasons these principles integrate well for computations with pixel-based images. Subsequently, an evaporation rate measured at the reference pixel could be scaled to all other pixels by multiplication with the value of $E_R$ at each pixel. These methods can be extended to surface variables (e.g. albedo, net radiation etc.) required to parameterise different evaporation models.

#### 3.3.2    Distributing surface variables using a normalised index of relative ratios

Methods described here assume spatial variations in surface variables driving net radiation (and driving evaporation) are near their maximum around solar noon. This is likely to be valid within 2 hours from the actual time of solar noon (Colaizzi et al., 2006). Net radiation is the major component needed to determine available energy for the conversion of water to an equivalent depth of water vapour. Net radiation is determined from the shortwave and longwave radiation components measured in the electromagnetic spectrum between approximately
0.3 - 4 μm and 4 to 14 μm respectively (Liang, 2004; Zoran and Stefan, 2006).

Traditionally, radiative terms needed for estimating evaporation are derived from satellite-based imagery, e.g. Landsat, AVHRR (Advanced Very High Resolution Radiometer) and MODIS (Moderate-resolution Imaging Spectroradiometer). However, satellite-based methods are continually limited by cloud contamination, varying spatial and temporal resolutions and sensor footprint mismatches. Under relatively clear skies it can be assumed
incoming shortwave and longwave radiation are uniform over large field areas. Very high resolution images taken





near the surface can then be used to derive the much more variable surface reflected shortwave and emitted longwave radiation components.

For example, normalized indexes for albedo, $\alpha_R$, emitted longwave, $L\uparrow_R$, and roughness height, $z_{oR}$, can be calculated at every pixel location within visible and thermal images following:

$$\alpha_R = \frac{\alpha_i}{\alpha_{ref}} \ , \tag{7}$$

$$L\uparrow_R = \frac{L\uparrow_i}{L\uparrow_{ref}} \ , \tag{8}$$

$$z_{o_R} = \frac{z_{o_i}}{z_{o_{ref}}} \ , \tag{9}$$

where subscript 'i' is individual values at each pixel and 'ref' is the value at the reference pixel (where $\alpha_R = 1$). Incoming shortwave ($K\downarrow$) and longwave radiation ($L\downarrow$) components can reasonably be assumed to be uniform over the field, so $\alpha_R$ and $L\uparrow_R$ can be further integrated to derive a normalised index of the midday net radiation, $Q*_R$, stated simply as:

$$Q*_R = \frac{Q*_i}{Q*_{ref}} = K\downarrow \ (1 - \alpha_R \ \alpha_{ref}) + L\downarrow - L\uparrow_R \ L\uparrow_{ref} \ . \tag{10}$$

Subsequently, a single measured value of mean daily net radiation taken at the reference pixel can be scaled across all other pixels via multiplication with the simple ratios derived for $Q*_R$. The next section illustrates the indexing method for deriving accurate estimates of albedo.

### 3.3.3 Normalised index method for albedo estimates from digital visible images

Surface albedo (α) represents a crucial radiation loss term for radiative transfer calculations and surface–atmosphere energy and mass exchanges (Sellers et al., 1997; Liang, 2000; Lucht et al., 2000; Roberts, 2001; Liang et al., 2003; Disney et al., 2004; Liang, 2004. Its calculation can be complex and is typically a major source of uncertainty (Yang et al., 2008).

In this case, a measured value of broadband albedo obtained at a reference pixel location was scaled to every other pixel within a high resolution visible image using Eq. (7). In 2007, digital photos were taken with a Canon Powershot A70 camera; max resolution 2048 x 1536 pixels, CCD (charge-coupled device) imager, DIGIC (Digital Imaging Core) processor. This resulted in very high resolution (< 1 m pixels) 'visible images' without cloud cover issues.

While digital cameras may not cover the full visible and near-infrared spectrum of advanced measurement sensors, the imaging techniques are still based on the same principles. Corripio (2004) demonstrated how accurate estimates of snow albedo could be obtained from digital images using a linear scaling technique applied to a measured albedo at a reference pixel location. A key step for our analysis was to transform an RGB digital photo to a single band 8-bit greyscale image with *DNs (*Digital Numbers) ranging from 0 – 255.

This resulted in higher *DN*s associated with more reflective surfaces (i.e brighter) and lower *DN*s less with less reflective surfaces (i.e. darker) in accordance with principles for calculating albedo. The resulting albedo map was aggregated to a pixel size of 5 m for a practical analysis and georectified to 100 GPS ground control points. The





study area included wetlands containing open water, so a simple Dark Object Subtraction (DOS) method was used to correct for potential atmospheric effects (Song et al., 2001; Liang, 2004).

Applying Eq. (11) made it possible to accurately estimate albedo at individual pixel locations, $\alpha_i$, from a measured broadband value taken at the reference location, $\alpha_{ref}$, and a normalised index for *DNs* calculated at each pixel:

$$\alpha^i = \alpha_{ref} \frac{DN_i}{DN_{ref}} , \qquad (11)$$

where $DN_i$ is the digital number of an individual pixel and $DN_{ref}$ is the reference pixel value.

### 3.3.4 Surface temperature (emitted longwave radiation)

For the case of surface temperatures the indexing method was not applied to generate distributed estimates because high resolution observations were directly obtained with a hand held Forward Looking Infrared (FLIR) ThermaCAM P20 imaging radiometer. The P20 used a Focal Plane Array, uncooled microbolometer, with a maximum image resolution of 320 x 240 pixels, a 24° by 18° field of view, and spatial resolution of 1.3 milliradians. The spectral range was 7.5 – 13 μm which is similar to traditional satellite sensors (e.g. Landsat, MODIS, and AVHRR 10 – 12.5 μm, and ASTER 8 – 12 μm).

A standard emissivity of 0.98 was assumed, and ambient air temperature / humidity and distance between the surface and camera detector were set based on observations.

The Stefan-Boltzmann equation was applied to transform surface temperatures into values of outgoing longwave radiation. For the flight height of approximately 1 km the FLIR produced a surface pixel resolution of < 3 m. The longwave radiation map was aggregated to 5 m resolution and then georectified using the resulting map of albedo estimates for reference.

### 3.3.5 Surface roughness

Surface roughnesss, $z_o$ is a critical component for aerodynamic calculations. For our purposes, $z_o$ was needed for calculting the "drying power" term in the G-D model. In this case, a classification map for $z_o$ values was derived from the 8-bit grayscale image used for estimating albedos. This was acheived based on knowledge of land cover heights at the site and and segmentation analysis using the IDRISI Kilimanjaro surface analysis tool.

Greyscale *DNs* were initially classified into 13 zones of similarity and a segmentation analysis was applied. The method computes a standard deviation for each pixel using a 3x3 moving window filter; the standard deviation and associated *DN* for each pixel is then sorted (low to high) and a bin range assigned; a class width tolerance was set for pixels having similar standard deviations and all values within a specified range were assigned to the same class. Where pixel values were outside the range, but class boundaries overlapped, a mid-point was determined and a new class is created.

The initial 13 classes were manually reclassified into three general classes (fallowed/cropped, grassed, and tree rings) based on a visual comparison of the original and classified images. Representative roughness heights, $z_o$ were then assigned to each class based on standard values for similar surface conditions (Brutsaert, 1982); 0.05 m for fallowed/cropped and 0.10 m for grassed areas.



The wetland tree rings were dense and relatively tall with heights varying from 3 m to 10 m. Brutsaert (1982) indicates expected roughness heights for vegetation ranging from 1 - 2 m and 8 - 10 m tall to be between 0.2 and 0.4 m respectively. For simplicity, a value of 0.40 m was assigned to all tree rings as they are assumed to have a simialr effect on turbulent fluxes. Areas between hillslopes where sharp changes in surface elements occured were also assigned a value of 0.40 m beacause turbulent fluxes might be ehnaced similar to a 'bluff' rough surface (Brutsaert, 1982).

### 3.4 Exploratory data analysis

Exploratory analysis of key surface variables and evaporation estimates was conducted using the 'R' software environment (Grunsky, 2002). Data analysis consisted of boxplot summaries using seven descriptive measures. Upper and lower limits of box whiskers defined the 75th and 25th quartiles (i.e. the interquartile range). Values 1.5 times larger or smaller than the interquartile range were indicated as open points and considered outliers. The median and mean values were defined by a solid line and a point respectively.

## 4 Results and discussion

Calculating detailed daily evaporation estimates and examining spatial scaling issues required the midday inputs and temporal scaling function to be computed first. Analysis of midday inputs are described in sections 4.1 – 4.4. Sections 4.5 and 4.6 discuss the sensitivity of the G-D model to the midday evaporation ratio and the temporal transfer function required for scaling a single measured value of mean daily radiation across a midday image. Sections 4.7 – 4.9 discuss the accuracy of the resulting evaporation estimates, the statistical distributions of the energy and aerodynamic components, and implications for improving larger scale evaporation estimates.

### 4.1 Surface reference meteorological parameters

At midday measured values of the incoming shortwave, $K\downarrow$ (835 W m$^{-2}$) and longwave, $L\downarrow$ (320 W m$^{-2}$) radiation and albedo were obtaiend from the CNR1. Both $K\downarrow$ and $L\downarrow$ were assumed to be uniform over the field given clouds and shadows were not a factor. Air temperature was also assumed to be constant based on similarities at three measurement locations (see Fig. 1).

Potential impacts of variations in humidity and wind speed were considered by examining G-D evaporation estimates derived from field data at two observation sites in 2006. During that period a portable EC/met station recorded observations over various land covers. A fixed station collected concurrent observations over a mixed grass upland. Meteorological observations from the fixed met station were intially used to estimate evaporation. The estimate was then recalculated after substituting observations of humidity and wind speed taken from the portable EC site.

Results showed no variaiton in the respective evaporation estimates (RMSE = 0.02 mm) derived from humidty and windspeed data taken from two different measurement sites. As a result, the humidity deficit and wind speed were assumed to be uniform over the area and set to the observed midday values of 1.09 kPa and 3 m s$^{-1}$ respectively.





### 4.2 Validation of albedo estimates

Angular measurements of broadband albedo from 0.3 – 3.0 μm with a hemispherical CNR1 directly account for bidirectional reflectance properties of the mixed grassed surface. Therefore, the observations satisfy bidirectional reflectance considerations related to albedo estimation techniques (Nicodemus et al., 1977; Lucht et al., 2000; Roberts, 2001). At midday on Aug 5, 2007 a measured reference albedo, $\alpha_{ref}$ = 0.153 was obtained from incoming and outgoing shortwave radiation over the mixed green grasses. The albedo map resulting from Eq. (11) and locations of reference and validation sites is shown in Fig. 2. Vegetation was similar at both locations and the scaled albedo estimate (0.164) agreed well with a measured value (0.167) at the validation site.

Validation of average and range of albedo estimates obtained for major land covers in the image is summarised in Tab. 1. Estimates of albedo from the image compared well with values expected for grasses, agricultural crops, deciduous trees, and gray bare soils reported in Brutsaert (1982). The 5 m resolution albedo map also highlighted key surface variations within the landscape that would be even more generalised if coarser data were used. For example, the resulting albedo map depicted distinct boundaries separating mixed grasses (MG), cultivated/crop area (C), sparsely vegetated, fallowed area (F), and wetland fringe vegetation (WL). Also, wetland extents and fringe vegetation were observable in areas surrounded by other vegetation types.

Figure 3 shows a sample of measured reflectance values from the grassed upland area. The spectral reflectance from the mixed grasses was virtually identical to the response for healthy (green) winter wheat (see Disney et al., 2004). For wheat, they concluded reflectance was directionally invariant and reasonably could be assumed to behave as a Lambertian surface (i.e. scattering light equally in all directions). The field measured spectral reflectance over the grassed surface was therefore treated similar to Landsat measured reflectance and divided into respective wavelengths for Landsat wavebands 1, 3, 4, 5, and 7.

An empirical linear approximation for narrow-to-broadband albedo conversion applicable to Landsat imagery (see Liang, 2000) was then applied to the field measured spectral reflectance data. This allowed for a direct comparison of albedo estimates and measurements along the sampling transect shown in Fig. 1. Due to the 4.5 m sampling distance some pixels contained a single value whilst others contained two, in which case the average value was taken. The root mean square error (RMSE) of the estimates and measured albedo values was approximately 3.5 % which is within an expected error of 2 % to 5 % for research purposes (Liang, 2004).

### 4.3 Validation of longwave radiation estimates

Figure 4 shows the resulting map of emitted longwave radiation, $L\uparrow$. The emitted radiation is estimated to be between 380 W m$^{-2}$ - 480 W m$^{-2}$. Lower values are expected to be attributed to increased water availability and higher values to water limited conditions due to reduced evaporative cooling. At two locations a comparison was made among the midday surface longwave measurements from the P20 camera, a narrow-beam Exergen infrared thermocouple (IRTC) radiometer and surface emitted longwave from the CNR1.

The P20 measurement compared well with the IRTC values with differences less than -12 W m$^{-2}$. Compared with the CNR1 values the differences where slightly larger at -30 W m$^{-2}$ but was still within 8 % error. The differences may be partly attributed to the absorption properties of water vapour (reducing signal) and the variable measurement footprints. Also, dust and heating of the CNR1 downward facing pyrgeometer can introduce a source



of error. Generally, the differences were small considering relative changes in midday surface net radiation and the magnitude of incoming radiation components is considerably larger.

### 4.4 Surface roughness height map

Figure 5 shows the resulting surface roughness height, $z_o$ map. A visual comparison with the longwave radiation map in Fig. 4 suggests the classification map is physically realistic. Notable variations in roughness and surface temperatures among the key land cover types were observed in both images particularly for areas covered by tree rings, fallow/cropped and mixed grasses and the distinct boundaries separating these cover types.

### 4.5 Sensitivity of the evaporation ratio to key variables at midday

The normalised indexes derived with Eq. (7 – 9) were applied here to examine relationships with the evaporation ratio. Figure 6 shows the expected physical behaviour within the G-D model. Only the actual range of values for the normalised indexes was considered so physical variations may be shown clearly. There is an apparent inverse linear relationship between $E_R$ and $\alpha_R$ and $L{\uparrow}_R$, and a slight non-linear relationship between $E_R$ and $z_{oR}$.

Consequently, a reduction in evaporation rates is expected where albedo, surface temperature and surface roughness tends to be higher. In the latter case, step changes in surface roughness appear to increase the relative drying power, $D$ but relative evaporation, $G$ decreases as a result of the inverse non-linear relationship. However, the impacts of relative changes among these variables is of greater interest here.

For example, a relative increase in $L{\uparrow}_R$ of 0.18 (or 18 %) reduced $E_R$ by 10 %. By comparison, an increase in $\alpha_R$ of 0.30 (or 30 %) reduced $E_R$ by only 5 %. In the case of surface roughness, an increase of 250 % was needed to reduce $E_R$ by 10 %. The increased sensitivity of $E_R$ to $L{\uparrow}_R$ indicates spatial variations in surface longwave radiation is an imprtant factor for estimating evaporation.

### 4.6 Temporal transfer function: normalised radiation ratio

Net radiation is known to vary dynamically on a sub-daily basis. Eq. (10) shows how a radiation ratio, $Q^*_R$ can be used as a temporal transfer function to scale estimates of mean daily net radiation to other locations. In order to scale the normalised net radiation ratio from a temporal "point" at midday to a mean daily value it was necessary to examine whether a stable proportionality exists between measured values at midday and mean daily net radiation.

Historical records were examined for a period from May 1 – Sept 1 at three Canadian prairie locations at similar latitudes (49° – 52.2°). The analysis included two field seasons at the SDNWA study site. Archived data was also obtained at two short grass prairie locations; an Ameriflux network site at Lethbridge, AB (1999-2004), and Kernen Farm located at Saskatoon, SK (1999-2000). Figure 7 shows a moderately strong relationship at each location; $r^2 = 0.54 - 0.6$. These results suggested the relationship between midday and daily net radiation might be stronger when midday net radiation exceeds 400 W m$^{-2}$, or likely on relatively cloud-free days.

Confirmation of an existing proportionality eliminated the need for an empirical relationship, and suggests daily net radiation, $Q^*_d$ can be scaled to all pixels across an image from a single measured reference value, $Q^*_{dref}$, as a function of the normalised ratio index for $Q^*_R$ at midday, stated here as:



$$Q_d^* = Q_{dref}^* \, Q_R^* \quad . \tag{12}$$

$Q^*_{dref}$ was assigned a measured reference value of 155 W m$^{-2}$ which was obtained from the CNR1. $Q^*_R$ was derived from Eq. (10) using the measured incoming shortwave and longwave radiation components, as well as the albedo and emitted longwave maps as input. The resulting mean daily net radiation map and a comparison of estimated and measured $Q^*_d$ at two locations is shown in Fig. 8. The validation site was equipped with a CNR1 and the second site maintained by Environment Canada was equipped with an NR Lite radiometer.

The distributed estimates of daily net radiation ranged from approximately 120 to 190 W m$^{-2}$. The $Q^*_d$ estimate at the validation site was 6 W m$^{-2}$ higher (4 % error) compared to the CNR1 measurement and 11 W m$^{-2}$ lower (8 % error) compared to the NR Lite observation. These results would indicate that accurate estimates of net radiation can be indexed across detailed midday images from a single measured value of mean daily net radiation. More importantly, accurate estimates of net radiation would be valuable for improving larger scale evaporation estimates.

### 4.7    Calculating direct estimates of actual evaporation with the G-D model

The mean daily net radiation map derived from Eq. (12) was supplied as input to the G-D model to estimate the mean daily "actual" evaporation at every land surface pixel. The soil heat flux was assumed to balance over the day. The reference air temperature, humidity and wind speed values (discussed earlier), and the surface roughness height map were used for calculating the aerodynamic terms in Eq. (2 – 6). An estimate of mean daily evaporation was then calculated at every pixel location using Eq. (1).

The resulting map of distributed actual evaporation estimates is shown in Fig. 9 and a visual inspection of the map shows a physically realistic pattern of evaporation for the range of land cover types. Areas where vegetation was sparse and soil conditions were likely drier showed lower rates of mean daily evaporation and higher rates were associated with more dense grasses (e.g. the upland area). The highest evaporation estimates were obtained where water availability is expected to be higher; e.g. among wetland fringes and some depressions where albedo and surface temperatures were lower, likely due to increased evaporative cooling.

The prevailing wind direction for the day ($U_{dir}$) was from a north-northwest direction. An evaporation estimate of 2.7 mm was obtained using Eq. (1) for pixels containing a brome grass surface immediately upwind of the EC station (approx. 100 m fetch). This model estimate compared well with the EC measured mean evaporative flux of 2.2 mm / day over the same brome grass area on the north-west side of Pond 1 as shown in Fig. 9. Observations were not available for tree rings (dominated by Aspen) in 2007 but estimates were compared against archived values from BOREAS data for an Old Aspen site from August, 1996. The G-D mean daily values in the order of 3 mm were reasonable compared to evaporation from tree rings reported by Hogg et al. (2000).

The results presented here are important for two reasons, 1.) the reliability of the G-D method for a complex landscape is clearly demonstrated, and Armstrong et al. (2008) also showed similar accuracy for daily and multi-day periods during the 2006 field season, and more importantly, 2.) methods applied here would be valid for scaling surface radiation components across remote sensing imagery obtained from satellite or aerial platforms such as UAVs.





### 4.8 Distributions of evaporation and driving surface variables

A key advantage of methods applied here is statistical distributions of evaporation and key driving factors are physically meaningful. Figure 10 indicates the frequency of evaporation estimates from images taken on Aug 5 were normally distributed. The G-D model calculated an average of 2.8 mm / day and the coefficient of variation (cv) was relatively small at 0.06.

The frequency of evaporation estimates attributed to the energy balance and aerodynamic components is shown in Fig. 11. Interestingly, these distributions were notably different from each other as a result of complex interactions among the driving variables. For the energy and aerodynamic components, the respective distribution means were 1.1 mm and 1.7 mm and the cv for each was 0.11 and 0.15.

Boxplot summaries of the distributions depict some notable differences. For example, Fig. 12 shows the albedo, $\alpha$ and surface temperature, Ts data were skewed in opposite directions; -0.83 for $\alpha$ and 0.36 for Ts. In the case of albedo, lower "outliers" were attributed to wetland vegetation and where surface water was not completely masked. Variability within the data was slightly larger for $\alpha$ (cv = 0.19) than for Ts (cv = 0.14). The range and standard deviation of evaporation estimates was much less than the net radiation when expressed in equivalent units (Fig. 13). However, they had similar variability, cv $\approx$ 0.07 which is relatively small compared to cv values for $\alpha$ and Ts.

A boxplot summary for distributed estimates of $G$ and general relationship with net radiation is shown in Fig. 14. The distribution of $G$ was not continuous which can be attributed to the step change in surface roughness height from 10 cm to 40 cm. Plots of $G$ against net radiation resulted in three distinct linear relationships. Increases in relative evaporation with net radiation in each case was due to the variability of surface state conditions across the field. More importantly, there was evidence of a non-linear, inverse relationship between the means of $G$ and net radiation across the roughness classes. This type of relationship might influence upscaled estimates of evaporation when calculated from larger scale averages of driving factors.

### 4.8.1 Variations within Roughness Classes

Variability within the roughness classes was considered further due to the relationship between $G$ and net radiation. Figure 15 shows boxplot summaries for both $\alpha$ and Ts. The respective distributions for $\alpha$ do not overlap due to the segmentation procedure used on the visible image. Albedo was considerably skewed in opposite directions for the 5 cm and 40 cm roughness heights; 1.4 and -1.6 respectively. In contrast, for the 10 cm roughness height the distribution appeared to be symmetrical (skew = 0.04) and showed less variability. This is not surprising given that this roughness class was comprised of various tall grass species.

By comparison, distributions of Ts overlapped considerably across the roughness classes. However, the interquartile ranges were noticeably offset with an increase in roughness class. Also, the skew notably shifted from left to right with increasing roughness; -0.32, 0.42, and 0.62. Figure 16 provides a summary of net radiation and evaporation within each class. When considered on appropriate scales the relative locations of the interquartile ranges, mean and median, skew, and "outliers" of these boxplots were similar within each roughness class. The data also exhibited lower variability (cv = 0.028 – 0.042) compared to the areal distributions of net radiation and evaporation (cv = 0.066) shown previously in Fig. 13.





In general, the relative shifts in mean α and Ts (Fig. 15) and net radiation (Fig. 16) among the roughness classes appeared to be linear. In contrast, Fig. 17 shows a non-linear behaviour for mean values of $G$ with increased roughness. The offset between the 5 cm and 10 cm roughness class heights was much smaller compared to the 10 cm and 40 cm roughness classes. As a result, potential increases in average estimates of $E$ associated with an increase in average energy availability would be offset by a non-linear reduction in average $G$.

### 4.9    Scaling implications

There is a potential for areal estimates of evaporation to vary depending on how upscaled estimates are calculated, which is examined here. average areal estimate of evaporation calculated from all pixel values was 2.8 mm/day. An areal estimate may also be calculated as the weighted average of evaporation from all roughness classes. This is similar to the mosaic approach which uses fractional land covers areas within land surface schemes. The mean daily evaporation rates for the 5 cm, 10 cm and 40 cm roughness classes were similar but also increased at each height as a result; 2.6, 2.8 and 3.0 mm/day respectively. Approximately 48% of the area was classified with a 10 cm roughness height, 30% of the area was classified as 5 cm and 22% of the area was classified as 40 cm. Therefore, a weighted areal evaporation ($E_{areal}$) can be calculated by:

$$E_{areal} = (0.30 * 2.6) + (0.48 * 2.8) + (0.22 * 3.0) = 2.78 \ mm/day \ . \tag{13}$$

In this case there was no difference in areal estimates obtained based on the distribution mean or a weighted mean based on fractional areas of the cover types within each roughness class. $E_{areal}$ was recalculated using Eq. (13) with different combinations of the fractional areas which only produced a minor difference of ± 0.1 mm. In other words, in order for there to be a larger difference between the areal estimates, greater variability is either required in the evaporation estimates distributed over the field or among the average rates for each roughness class.

The areal estimate of 2.8 mm/day was considered against point measurements and estimates on the west side of pond 1. For this case $E_{areal}$ was 0.6 mm higher than the 2.2 mm/day measured over the grassed location. $E_{areal}$ was also 0.3 mm higher than the G-D model estimate of 2.5 mm/day obtained from station meteorological observations including net radiation, air temperature, humidity, and windspeed. $E_{areal}$ was only 0.1 mm higher than the areal estimate of 2.7 mm/day obtained from the 2000 m² grassed area upwind of the eddy covariance station. Variations among the estimates and measurements are relatively small, but not surprising given the differences in calculation techniques and associated footprint scales.

An upscaled areal mean estimate of evaporation can also be obtained from mean values of the key factors driving the energy and aerodynamic terms. The general form of Eq. (1) can be rewritten as:

$$E = \frac{\Delta_G Q^*}{\Delta_G + \gamma} + \frac{\gamma_G E_A}{\Delta_G + \gamma} \ . \tag{14}$$

For the entire area and also each roughness class, average values of the driving factors were derived and evaporation estimates were recalculated using Eq. (14). The relative evaporative contributions attributed to the energy and aerodynamic terms are provided in Tab. 2. For the different roughness classes the range of evaporation estimates attributed to E_energy was only 0.2 mm/day and nearly 0.7 mm/day for E_aero, and the difference in total evaporation, E_total was only 0.5 mm/day. A bias toward larger evaporation estimates might be expected





given the increase in energy availability and enhanced turbulence with an increase in roughness height. However, any potential bias seems to be offset by the non-linear, inverse relationship between mean values of $G$ and $Q^*$, and also $G$ and $E_A$.

Table 2 also compares evaporation estimates calculated from only mean values of $G$ and $Q^*$ and $E_A$ to the expected rates for each roughness class. "Expected" rates were calculated from the mean value for all pixels assigned within each roughness class. Evaporation rates derived from the mean values alone were between 0.14 mm and 0.2 mm less than the expected averages for each roughness classes. Upscaling the driving factors to the entire area also had no impact on the estimate as the difference was just 0.1 mm.

### 4.10 Examining covariance among key variables

Whether evaporation estimates might be influenced by a covariance between driving factors was also examined. The Pearson correlation coefficient, r was used to evaluate correlations among the driving factors distributed over the field area. By definition, Pearson's correlation is the ratio of the covariance (the numerator) between two variables normalised by the product of their standard deviations as follows:

$$r = \frac{\sum \frac{(X_i - \bar{X})(Y_i - \bar{Y})}{n}}{\sqrt{\frac{\sum(X_i - \bar{X})^2}{n}} \sqrt{\frac{\sum(Y_i - \bar{Y})^2}{n}}} \ , \tag{15}$$

where Xi and Yi are the respective values of the variables, the overbar denotes the mean value and n is the number of pairs. A strong correlation between two variables might suggest the existence of covariance that could influence upscaled estimates of evaporation. Given the roughness classes used represent discrete data, and a lack of detailed meteorological data, further evaluation in relation to climate factors would be less meaningful. So only an examination of the factors driving the energy term can be considered here.

In this case, a potential covariance between G (dimensionless) and Q* (expressed in mm/day) can be considered directly because they are multiplied together. This is also true for G and EA except that EA had only three discrete values for roughness class. In both cases the covariance is expected to be negative due to the inverse relationship among means. By rearranging Eq. (15) the covariance can be obtained by multiplying the correlation coefficient and the product of the standard deviations of Q* (0.34 mm/day) and G (0.021 mm/day) depicted in Fig. 13 and 14. The correlation between Q* and G over the field area produced a coefficient, r = -0.67.

When multiplied in series (r = -0.67)*0.34*0.02 this resulted in a covariance of approximately -0.0046 mm/day. This result suggests spatial interactions between Q* and G would have no further statistical influence on upscaled evaporation estimates. Unfortunately, no comment can be made regarding covariance between G and the turbulent flux component in the G-D model. Such analysis would require more detailed observations of air temperature, humidity and wind speed, and likely a more sensitive combination model. However, combined interactions within the G-D model appear to produce reliable average estimates of evaporation over larger scale areas.





## 5    Summary and conclusions

This study examined spatial associations and physical interactions among key surface variables driving actual evaporation estimates, and impacts of their variations on upscaled estimates. The methods applied demonstrate how a measured reference value of albedo and mean daily net radiation can be scaled accurately across a large

5    field area. This was achieved by computing a normalised index of relative ratios using highly detailed midday visible and thermal images. At two validation sites estimates of daily net radiation showed good agreement with measured values to within 4% and 8% error. Estimates of mean daily actual evaporation were calculated at 5 m resolution with the G-D model.

The "upwind" daily evaporation rate was 2.7 mm for Aug 5, 2007 which was only 0.5 mm larger than the 2.2

10   mm observed using eddy covariance. Differences in areal evaporation estimates were relatively small regardless of whether, 1. Representative average values of driving factors were used to parameterise the model, or 2. An average evaporation rate was derived from detailed estimates across the field. There was no evidence of a covariance between the spatial distributions of net radiation and $G$, and therefore, no correction factor could be identified for improving upscaled evaporation estimates.

Offsetting interactions between the relative evaporation term and key surface variables effectively reduced the spatial variability of evaporation estimates. The methods applied here could generate useful diagnostic information at other locations and potentially over much larger areas. In particular, these methods would be valuable for research or other applications where detailed images can be obtained from digital cameras and sensors mounted on UAVs. The methods applied here may also be instructive for improved upscaling of evaporation estimates

where more traditional remote sensing or climate modelling methods are used.

## Acknowledgements

Funding for this research project was provided by the Drought Research Initiative (DRI) and the Canadian Research Chairs (CRC) programme.

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





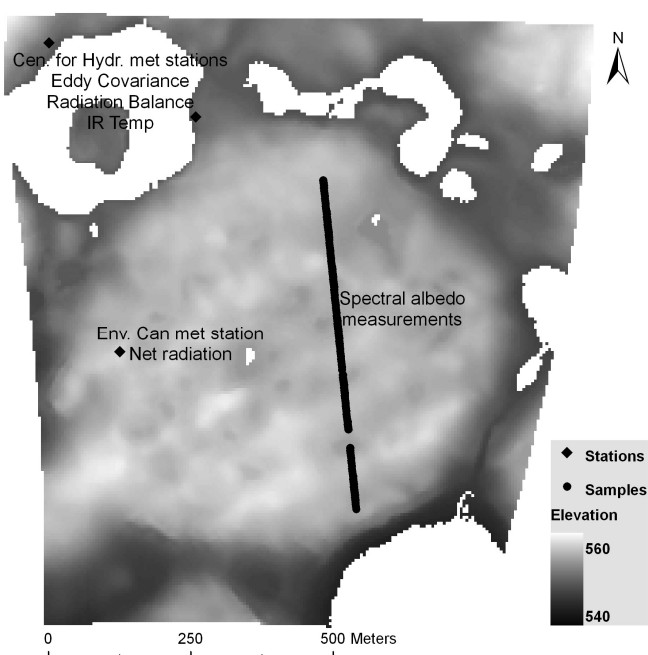

**Figure 1: Land surface elevation map and locations of measurement sites for eddy covariance and micrometeorological observations, and sample points for spectral albedo along an existing transect with 4.5 m spacing.**





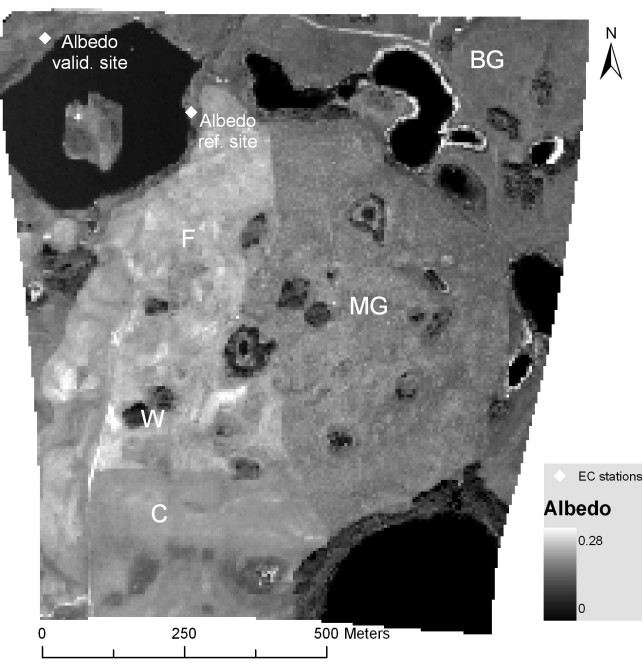

**Figure 2: Albedo map (5 m resolution) derived from visible image taken at midday. Also shows location of reference and validation sites, letter codes indicate major land cover types: fallowed (F), mixed grass (MG), brome grass (BG), cultivated (C), and wetlands (W).**



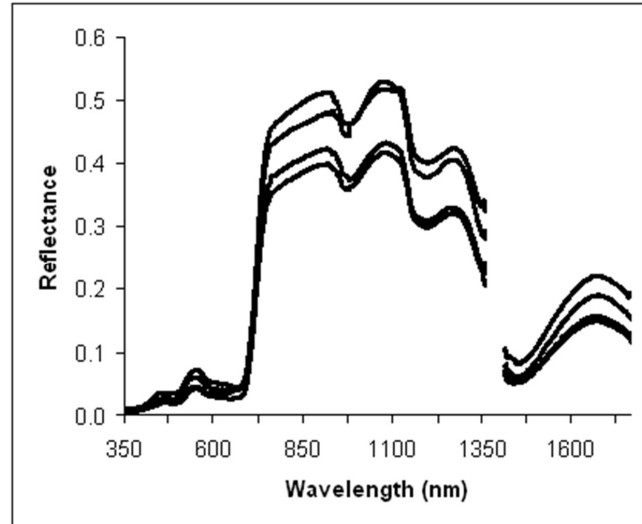

**Figure 3: Reflectance spectra collected at four sample points over mixed grassland vegetation at the upland area on Aug 5, 2007. Reflectance values affected by noise at corresponding wavelengths were removed.**




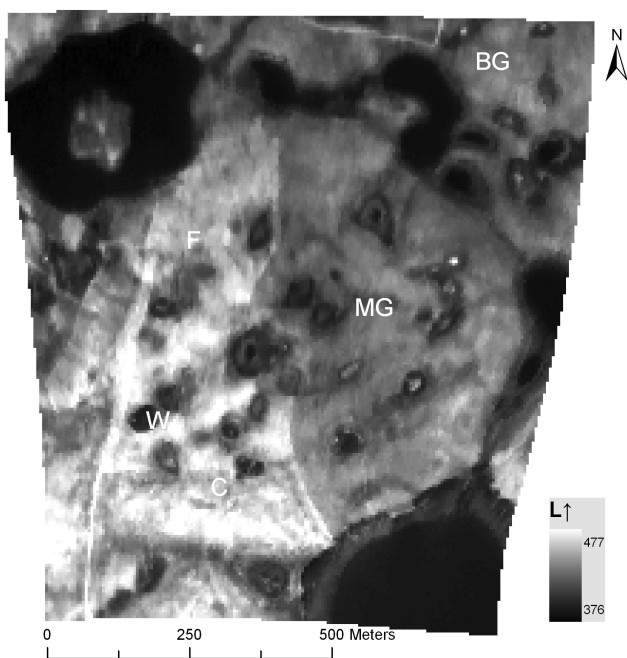

**Figure 4: Surface emitted longwave radiation (W m$^{-2}$) map (5 m resolution) derived from a thermal image taken at midday.**



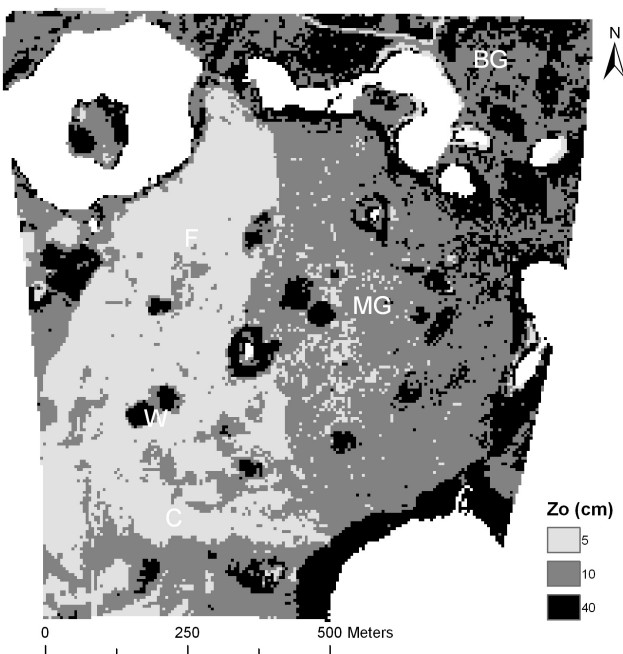

**Figure 5: Classification map of aerodynamic surface roughness heights derived from a visible image taken at midday and typical values found in Brutsaert (1982).**





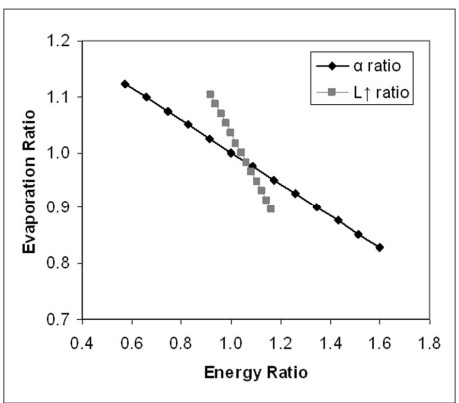 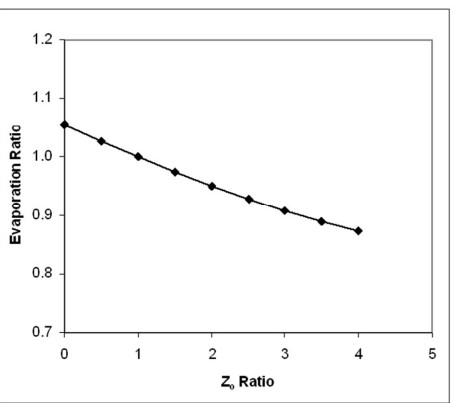

**Figure 6: Sensitivity of the evaporation ratio to key inputs at midday. The measured range of input values is shown to demonstrate potential variation in this case study.**





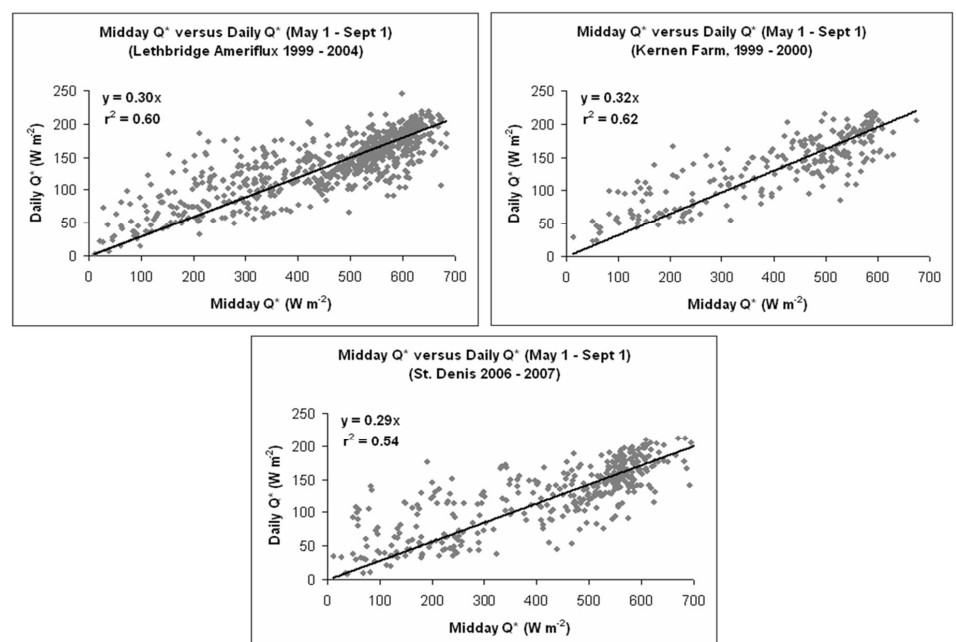

**Figure 7: Relationship between the midday and mean daily net radiation for a range of years at two Canadian Prairie sites and one Parkland site for the period May 1 through September 1.**





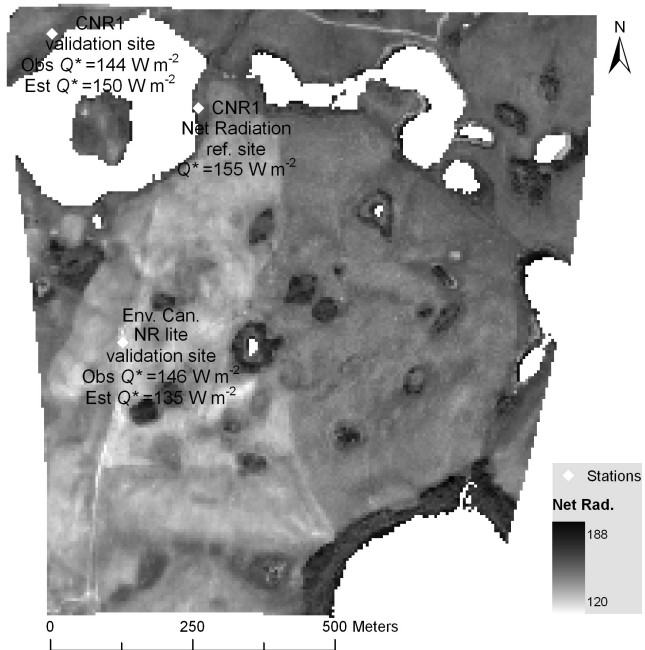

**Figure 8: Resulting input map of mean daily net radiation derived from the normalised index of midday net radiation and a single reference value of mean daily net radiation (155 W m$^{-2}$). Also shows location of validation sites for comparing measured and estimated values of mean daily net radiation.**



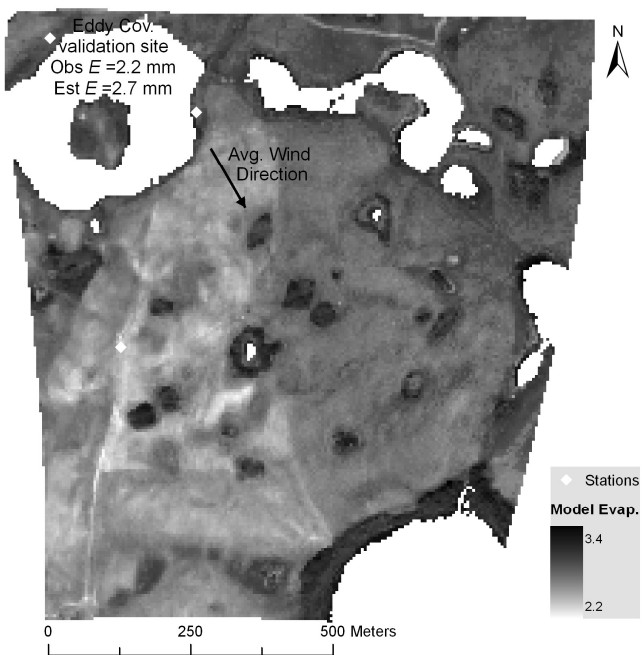

**Figure 9: Map of distributed estimates of mean daily evaporation at 5 m pixel resolution.**





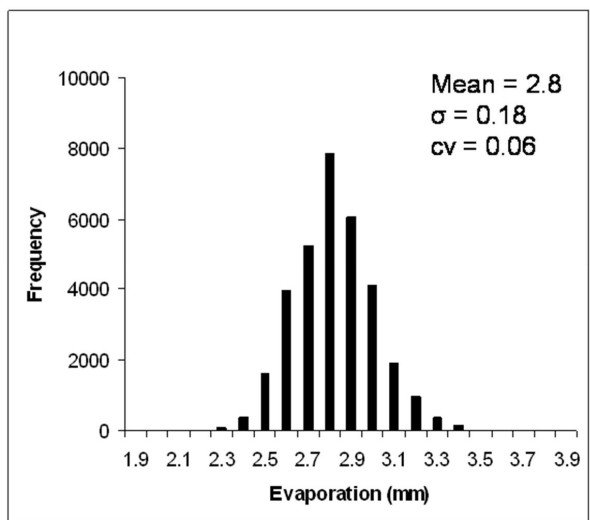

**Figure 10: Distribution of daily evaporation estimates over the field area.**





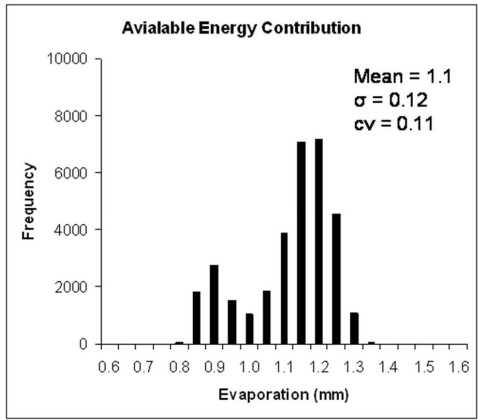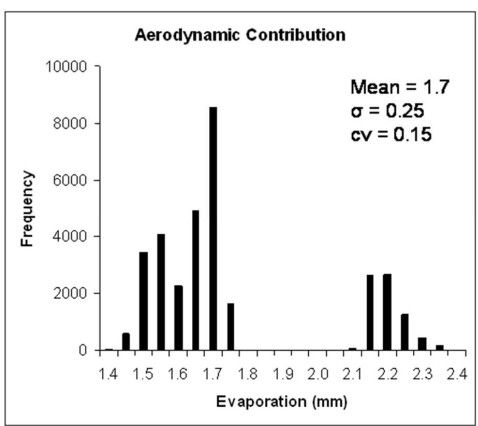

**Figure 11: Relative contributions of evaporation for the energy and aerodynamic terms.**



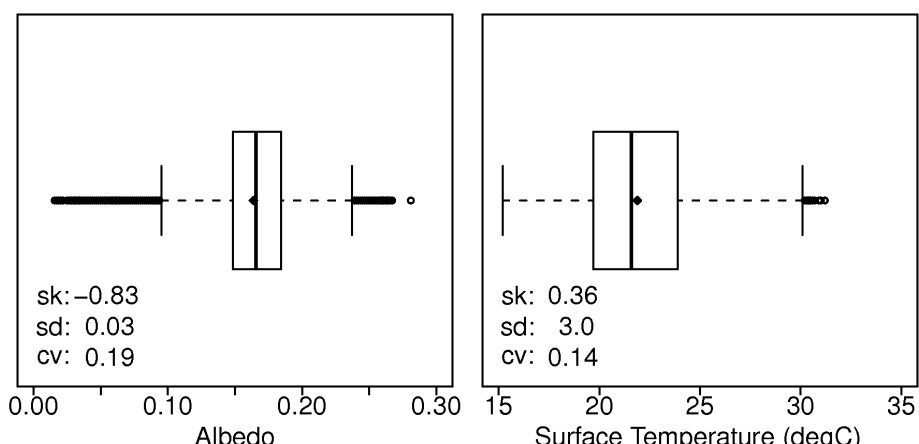

**Figure 12: Distributions of albedo and surface temperatures.**




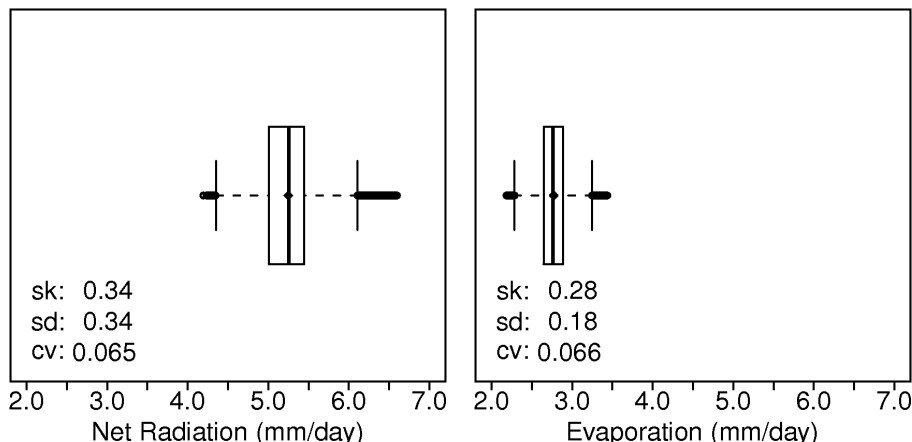

**Figure 13: Distributions for mean daily net radiation in equivalent units of evaporation, and the mean daily evaporation estimate.**





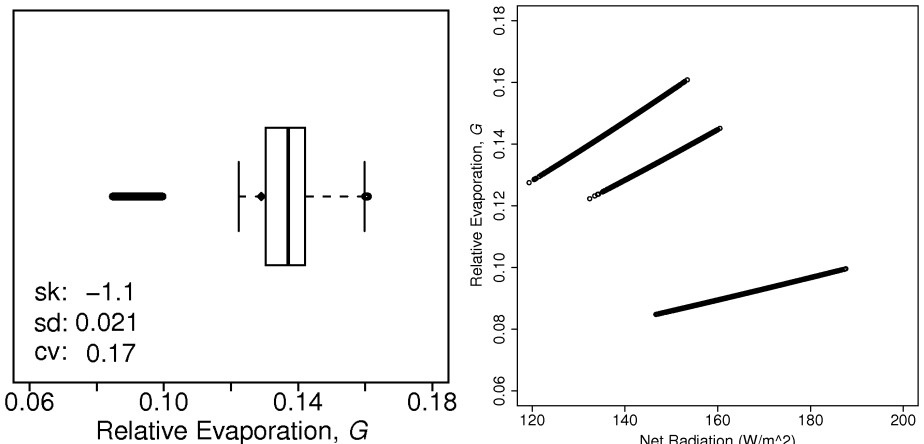

**Figure 14: Distribution for relative evaporation, G and relationship between G and net radiation within each roughness class.**




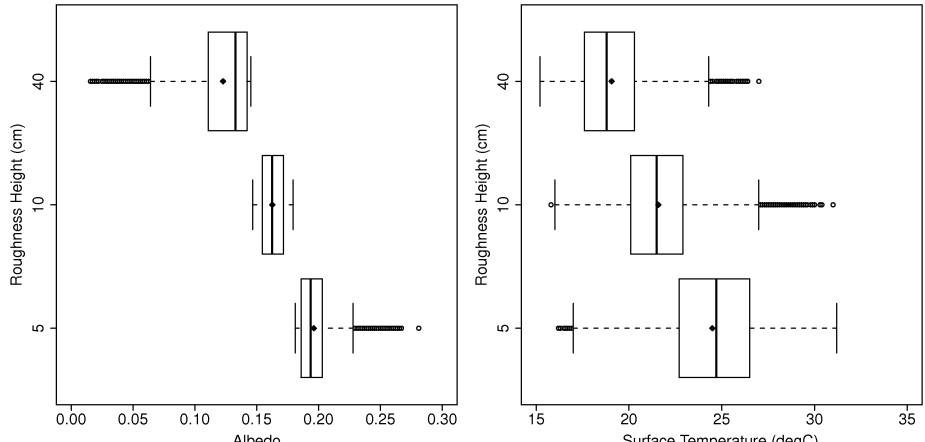

**Figure 15: Distribution of albedo and surface temperature within each roughness class.**





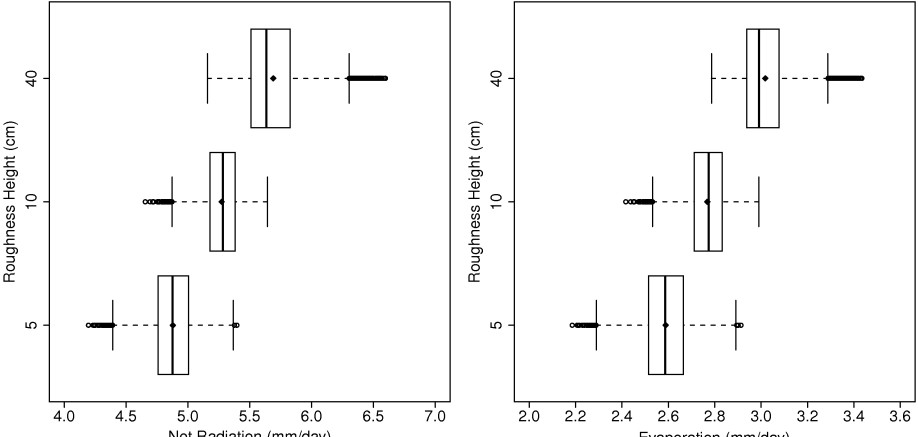

**Figure 16: Distribution of evaporation and net radiation within each roughness class.**




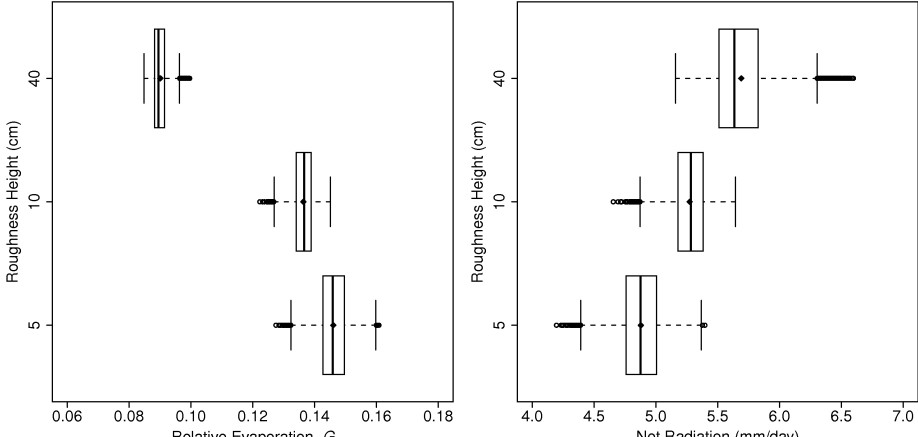

**Figure 17: Non-linear decline in mean relative evaporation and linear increase in mean net radiation with increasing roughness height.**





**Table 1: Approximate mean values and ranges of albedo for the major landcover types.**

| Land cover | Mean | Range |
|---|---|---|
| Wetland vegetation (W) | 0.11 | 0.05 – 0.16 |
| Brome grass (BG) | 0.15 | 0.13 – 0.17 |
| Mixed grass (MG) | 0.17 | 0.15 – 0.19 |
| Cultivated (C) | 0.18 | 0.17 – 0.20 |
| Fallowed (F) | 0.20 | 0.17 – 0.23 |





**Table 2: Areal evaporation estimates within each roughness class from G-D model and for entire area based on mean values. E_energy, E_aero are the contributions from the energy and aerodynamic components and E_total is the combined total. The mean value of the distributed estimates is given by "Expected" and the difference between the total and expected is given by "Diff".**

| $Z_0$ | $\Delta$ | G | $Q^*$ | $\gamma$ | $E_A$ | D | $E$_energy | $E$_aero | $E$_total | Expected | Diff |
|---|---|---|---|---|---|---|---|---|---|---|---|
| cm | kPa | | mm/day | kPa | mm/day | | mm/day | mm/day | mm/day | mm/day | mm |
| 5 | 0.134 | 0.132 | 4.88 | 0.063 | 12.99 | 0.73 | 1.07 | 1.34 | 2.40 | 2.59 | -0.18 |
| 10 | 0.134 | 0.124 | 5.27 | 0.063 | 15.13 | 0.74 | 1.10 | 1.48 | 2.58 | 2.77 | -0.19 |
| 40 | 0.134 | 0.085 | 5.69 | 0.063 | 27.97 | 0.83 | 0.87 | 2.01 | 2.88 | 3.02 | -0.14 |
| $E_{areal}$ | 0.134 | 0.113 | 5.28 | 0.063 | 18.70 | 0.77 | 1.03 | 1.71 | 2.73 | 2.77 | -0.03 |

