# Peer review of "Spatial variability of mean daily estimates of actual evaporation from remotely sensed imagery and surface reference data"

_Hydrology and Earth System Sciences, 2018_

## Referee Comment (RC1) · Anonymous Referee #1 · 22 Feb 2019

The novel approach to spatially scaling evaporative fluxes could be a significant contribution to HESS and quite valuable for some readers. It is very exciting to think of scaling point measurements to the widely available and affordable remote sensing data. However, the manuscript requires some restructuring for clarity and a much deeper discussion on potential implications and limitations. The model is clearly presented, and with some minor clarifications could be easily reproduced. Some figures could be combined or eliminated. The biggest weakness is that the discussion and conclusion lack a detailed assessment of the uncertainty in the model, including what regions it might have performed poorly in and more general speculation on its applicability outside the study area.

Specific comments:

Fundamentally the authors need to discuss the implications of assuming the ground heat flux is negligible while at the same time utilizing differences in surface temperature to derive variations in the upwelling longwave radiation (and thus available energy for evaporation). Perhaps this is a minor error, but that is unlikely given the gap in explained energy shown in Figure 13. If all that unaccounted for energy is partitioned into sensible heat it would require Bowen ratios of around $\sim 0.9$, which is probably too high for a grassland. If the unaccounted for energy is turned into kinetic energy, it will contradict other assumptions in the model. This project would benefit from a discussion about the implications of these assumptions and speculation of their magnitude using the energy partitions observed at the EC system that day.

More information needs to be provided about the EC data used to validate the model. With only one day of measurement, it is critical that it is a top quality flux (or at least specify any issues clearly). At a minimum standard quality control metrics should be reported for the flux (e.g. a 0 flagged flux by the methods of Mauder and Foken 2004), and it should be clearly stated that the measurements were not made in the wake of the tower at any point during the day. The authors should report if any gap filling techniques were required to derive the daily flux and which methods were used to calculate the spectral corrections and density corrections. Furthermore, the 100 m EC fetch should be overlaid onto the map, or clearly outlined how it was defined in the text. Is it simply a linear transect, an ellipse, or derived from a footprint model? Since this is the primary validation, it should be clear what segment of modelled flux from the surface is being compared to the EC fluxes.

A discussion on the uncertainty is lacking. Are any of the crops C4 plants? Could that be a problem in other systems? How likely is it that wind speed and turbulent energy is consistent across all surface classes? Abrupt changes in surface, from smooth open water, to forest, to hill slopes, does not simply translate to a greater roughness (as stated on Page 8, Ln 4/5). Internal boundary layers and small pressure cells produce

microclimates as a result of divergent or convergent flow at abrupt roughness and topographic changes. These microclimates could increase air flow through the canopy (thus reducing roughness height and surface resistance) or decrease flow within the canopy (increasing roughness height and surface resistance) and drastically change Ea, which will depend on the direction of the wind at the time. This may be trivial on the overall estimates presented in this paper, or it could be significant (depending on the surface heterogeneity and wind variability) but more evidence that the authors have considered this would help a reader understand where and when the model is applicable.

Figure 15 shows the only true variability in the model outside those directly measured at a point location (since Zo is classified from the same DN that defines the albedo). It seems appropriate to start the discussion with the variability in DN (or alpha) and Ts (or upwelling Longwave), not conclude with it since all other results can relate to this variability in surface parameters. Specifically, the authors should address how the variability in DN and surface temperature impact estimates for E.

On page 14, line 26, the covariance between Q* and G was stated to be very small. However, G is 100% dependent on Q*. Does this unorthodox computation of covariance appear greater when calculated within each roughness class? More importantly, what is the covariance between your remotely sensed variables (DN and Ts)? Could this be a problem if these are highly correlated?

Technical Corrections:

Reference the company that makes the software/toolbox used to partition the surface.

Some figures are barely discussed in the text, making the reader wonder how relevant all 17 figures are. For instance, Figure 13 was only discussed in one sentence and briefly referenced in another. Figure 12 is referenced once, then completely described in text such that the figure added no value.

Add jitter to the boxplots outliers so the density of outliers can be assessed.

You don't need both CV and SD since they provide redundant information when the mean is available. Since you have such high N, you technically don't need either because the interquartile range is adequately defined and it is a better descriptor of the variability when the data is skewed.

Remove the coefficient of variation of the aerodynamic component (page 12, Ln 9). It looks like the distribution is highly bimodal and you don't refer to it anywhere.

Vegetation types are not clearly defined. Stick to the same discerptions throughout, and be sure all are labelled on the map. "Tree rings" suggest a site used for dendrochronological analysis, did you mean a small corpse of trees surrounding the wetlands (a ring of trees)? If so this is not clear.

Numerous typos were found, please edit for clarity. This is not a comprehensive list, but these jumped out while reading: Pg. 15 Ln 11, Pg. 13 Ln 8, Pg. 7 Ln 15, Pg. 6 Ln 19, and Equation 5.

The paragraph starting on line 20 of page 4 is difficult to follow. Are there three radiation validation sites, or two? I had to check the results to understand your methods. Please clarify.

Don't describe the visual difference, simply state if differences are significant or not (e.g. Page 12 Line 34/35). If it is not normal, just use a non-parametric test (something simple like Wilcoxon).

A roughness length of 0.4 m seems short for a dense stand of 10 m tall trees; please justify.

Title of section 3.4 is misleading. Is this exploratory analysis or an analysis of the variability within the model?

600, 2018.

---

## Referee Comment (RC2) · Anonymous Referee #2 · 20 Mar 2019

This study illustrates methods to calculate normalized indices of surface characteristics to estimate evaporation and its spatial variation using the Granger and Gray model. Spatial variability of evaporation in this parkland area was low. Although aspects of this study were interesting, problems with clarity, organization and very limited discussion made it difficult to assess the contributions of this work and its conclusions.

Specific comments: The introduction/background did not clearly describe the study objectives and how it was new and would advance an understanding of the topic. More specific and well-defined research objectives including imagery resolution/scale detail and hypotheses would be helpful. Last sentence of section 1 detracts. Perhaps move

into discussion.

Methods: Is it necessary to repeat the normalized index equation for each variable? It would be helpful to have all the reference values for the normalized variables in one place as well.

The eddy covariance measurements of evaporation and calculation methods are not well described. Are energy closure corrections or other corrections applied? There is a 2006 EC study mentioned in the results that is not described in the methods nor referenced. If that study is important, then results should be given to support the conclusions that are drawn (perhaps in an appendix). At the moment, it seems to raise more questions than it answers.

Figure 1 was not helpful for context. Perhaps a colour image of the region or instead, include the station and sampling points on Figure 2. If I understand correctly, the ∼100 m upwind fetch of the EC station is not shown on the map. Why not? Ponds are not labelled that I can see but are mentioned in the text, pg 11.

Results/discussion: There is a mix of methods and results in the results/discussion section. Clarity would be improved if these were better separated. For example, the corrections applied to the field measurements of broadband albedo to allow for comparison with narrow-band albedo could be included in the methods.

Was the study day cloud free? If so, then yes, daily and midday net radiation would scale very well. There is a large body of literature on net radiation models that could better constrain the uncertainty with which midday net radiation is useful for estimating daily values. Figure 7 relationships may not be useful in the long-term nor applicable elsewhere.

What is the basis for evaluating the evaporation measurements in Section 4.7? A difference of 0.5 mm/day is not insignificant.

The purpose of describing the distribution of the variables (Section 4.8 and Figures 10 -

14) and the covariance analysis (Section 4.10) could be made clearer. As noted above, specific objectives or hypotheses could be useful. The differences among roughness 'classes' in Section 4.8.1 is interesting but there is limited discussion.

Since the radiative/energy and aerodynamic terms are discussed earlier in the results (Fig 11), perhaps present equation 14 instead of equation 1 at the beginning of the paper.

There are a number of typographical errors throughout. Ensure all acronyms are defined on first use. The number of figures could be reduced and quality of most map Figures could be improved.

––––––––––––––––––––––––––

---

## Author Comment (AC1) · 2 May 2019

We are grateful for the comments from RC2. The following responses are intended to address the range of comments given by the reviewer regarding objectives, organisation, methods, and results and discussion.

The study objectives and potential for advancing understanding for examining improved methods of upscaling evaporation estimates has now been clarified in the introduction / background section. The ratio indexing methods represent a new way of scaling point measurements across large fields for evaporation modelling.

[Figure]

Portions of the text have been reorganised and edited based on the comments regarding clarity and organisation. Specifically, the last sentence of section 1 has been better integrated earlier in the introduction/background. Regarding the mix of methods and results, text related to the methods has now been worked into the methods section.

Regarding comments related to the Methods: Repetition of the normalised index equations, i.e. equations 7-9 are referred to later in section 4.5 and Equation 10 shows the general integration for calculating net radiation – it may be preferable to leave these as is for clarity.

The reference parameters section and relevant parameters have been moved into the methods section for clarity.

A discussion of the Eddy covariance measurements and corrections has been clarified in the field observation section 3.2 and further in a new section (4.10) in relation to the modelling uncertainty. The confusion regarding the 2006 EC study has been addressed in the text – the data collected was referenced by the Armstrong et al., 2008 study cited in the manuscript, but the year 2006 refers to when the data was collected, which has been clarified in section 3.1. The relevant text related to the methods was moved section 3.3.

In the interest of reducing costs of publication, using greyscale images/figures is preferred for this manuscript. Figure 1 has been replaced with an actual greyscale photo the region taken during the study flight on August 5 2007. The albedo sampling points from Figure 1 were moved to figure 2. Replacement of Figure 1 provides clearer context so specific references to ponds have been removed as they can be seen clearly in the photo.

The text referring to upwind fetch has been clarified to indicate that 80% of the upwind contribution comes from 100 m upwind of the EC station, based on the cumulative flux calculation with the footprint model of Scheupp et al, 1990. This is along a similar linear transect used for averaging the G-D model estimates upwind of the EC station which

has now been added to Figure 9.

Regarding comments related to the Results and Discussion: The text which describes the corrections applied to the field measurements of broadband albedo have now been moved into the methods section 3.4.3 under the methods describing the derivation of the normalised index for albedo.

It has been more clearly stated the study was relatively cloud free – the data shows just two 15 min periods later in the day when clouds passed over.

With respect to the Figure 7 relationships - the regression equation was removed from the figure. The r-square was left as it simply reflects the validity of assuming net radiation at midday can be used for temporal scaling of mean daily net radiation which appears to be more stable under clear sky conditions.

The basis for comparing the estimates and measured evaporation in section 4.7, now section 4.6, has been stated in terms of % error in the overestimate to be more relevant.

The purpose of section 4.8 and 4.8.1 (now 4.7. and 4.71.) have been edited to improve clarity and several figures have been combined into a single figure for more clarity and relevance to the discussion.

In the interest for avoiding confusion on the general notation of the evaporation equation (Eq. 1) and rearrangement to obtain contributions from the individual components (Eq. 14) it may be preferred to keep the two equations separate, but if crucial this can be changed.

Typographical errors have been re-checked and edited where found. It is also noted that British spellings are also being applied throughout.

The number of figures has been reduced from 17 to 12 by combining figures 10 – 11 (now fig 10), 12 – 14 (now fig 11), and 15 – 17 (now fig 12).
* * *
600, 2018.

---

## Author Comment (AC2) · 7 May 2019

We are grateful for the comments from RC1. The following responses are intended to discuss key edits which have been made to the manuscript based on the range of comments given by the reviewer regarding: restructuring of the manuscript, discussion on potential implications and limitations, energy flux and EC data and modelling uncertainty.

Regarding comments related to the EC data:

More information has been given regarding the EC measurements in the field observation section 3.2. For example, fluxes reported as 15 min averages, data filtering was used but there was no bad data or spikes observed on the day. All flux samples were accounted for and no gap filling was required. Use of the planar-fit axis rotation method to correct the latent heat flux (and sensible heat flux) measurements was also previously referred to in section 3.2. Given the corrections applied and no missing data or spikes were observed, the fluxes were considered to be of good quality for the case study. Unfortunately, post-processing of the raw data is not possible at this stage to generate quality flags according to the methods of Mauder and Foken 2004.

Partitioning of the energy fluxes is now discussed in section 4.6 where a comparison is made between the estimated evaporation rate from a linear transect upwind of the EC station and the measured EC flux. The linear transect has been overlayed on the map in Figure 9. The text referring to the flux contributions from the upwind fetch has been clarified. Specifically it has been noted that 80% of the upwind contribution is expected to come from within 100 m of the EC station, based on a cumulative flux calculation with the model of Scheupp et al, 1990. This is along a similar linear transect used for averaging the G-D model estimates upwind of the EC station which has now been clarified in the text.

In terms of the flux components and ratio of energy balance closure, the fluxes were specified as mean daily values in W/m2: $(LE + HE) / (Q^* - Qg) = (63 + 55) / (144 - 2)$ which gives a closure of 83%. The Bowen Ratio of 0.87 is reasonable in this semi-arid landscape due to drying of the upwind grass surface and reduced photosynthesis of the grasses as anthesis is typically in mid-late June. Uncertainty in over estimating evaporation due to neglect of the ground heat flux and possibly under measured fluxes is now referred to in section 4.6.

Regarding comments related to modelling assumptions and uncertainty:

Uncertainty associated with the modelling assumptions has been partly addressed by moving the surface reference parameter section (now 3.3) into the methods and text

modified to provide clarity regarding the assumptions applied. A new section (4.10) was added to further discuss the general uncertainty of the methods applied. This includes discussion related to regions where the model may perform more poorly due to neglect of the ground heat flux or wind speeds may vary due to the changes in the roughness of the surface elements.

The relatively small magnitude of the ground heat fluxes under grass surfaces with good cover at two measurement sites has now been discussed in section 3.3 and is referred to again in section 4.10. A focus of this study was on the potential to scale measured values of driving energy factors across the larger field based on the observable surface properties directly related to the net radiation. Which is not practical in this case for the ground heat flux without further information on the land cover in each pixel and application of a numerical model. Estimates could be produced at every pixel through complex radiative modelling but the uncertainty may be larger than the error of the estimates reported for the net radiation estimates. It is not uncommon to neglect the ground heat flux but implications are discussed in section 4.10.

To the best of our knowledge there were no C4 plants at the study location – all grasses/crops were cool season C3 types. Uncertainty related to prior developments of the vapour transfer function for different surface types (including C3 plants) and possible requirement of new equations for C4 plants is now noted in section 4.10.

Regarding comments related to the wind speed and turbulent energy applicable to the estimating the drying power of the air, EA:

The discussion regarding development of the surface roughness length map in section 3.4.5 has been improved for clarity and to provide evidence of considerations for potential impacts of changes in roughness and wind speed. As was stated in section 3.4.5 representative roughness lengths were selected based on reported values from Brutsaert (1982) for similar types of surfaces elements and heights for vegetation ranging between 3 – 10 m. Regarding restructuring and combining images: The resulting

maps of albedo and Ts and the validation discussion (section 4. 1 and 4.2) are introduced earlier in the result and discussion section. Implications of relative variations in albedo and TS for estimating the net radiation and final estimates of E are discussed briefly in each of those sections. A major restructuring was done to combine several images which reduced the number of figures from 17 to 12. The manuscript has been modified in section 4.7 and 4.7.1 and graphics have been combined so as to be more appropriate to discuss variations in the underlying distributions of driving of factors and how they relate to the net radiation and impact final estimates of E. For example the underlying variability in albedo and TS appears to be much larger than the resulting variability of net radiation and E, due to the interaction of the relative evaporation, G term. Previous figures for the frequency distributions for evaporation estimates and relative contributions have been combined in to Fig. 10. Previous figures 12 – 14 have been combined into Fig 11 and previous figures 15 – 17 have been combined into Fig 12.

Previous figures 2 and 3 have been swapped as the discussion for figure 2 related to albedo point sampling and conversion to broad-band was moved into the methods section.

Regarding comments on covariance:

The text has been modified to clarify the impact of the relationship between relative evaporation G and net radiation. G is a non-linear function of the relative drying power D which is a function of the drying power of the air, EA and also the available energy. EA is dependent on the surface roughness, wind speed and humidity deficit. The non-linear inverse relationship between net radiation and G is more clearly shown and discussed via figure 12. The impact of this relationship for this study generally results in higher values of G associated with lower values of net radiation and much lower values of G associated with higher values of net radiation. The resulting interaction produced a very small covariance.

The covariance was found to be even smaller when computed within each roughness class.

Albedo estimates which were based on the DN index showed a correlation of 0.67 with TS, which is reasonably high. However, the computed covariance was only 0.06, suggesting that this may not be an issue.

Regarding technical corrections:

Idrisi software used to segment the surface now referenced, and company referenced for post-processing done with Matlab software.

As indicated above several figures have been combined and discussed more appropriately.

The boxplots have been modified to remove the redundant information and the large number of data points have been overlayed and plotted with jitter.

Removed the coefficient of variation for the aerodynamic component as it appears bimodal.

Vegetation types have been more clearly defined – also figure 1 was replaced with a photo of the study region which provides more context for reference.

Tree rings clarified to narrow rings of trees.

The manuscript has been cleaned up to address typos.

The number of validation sites for net radiation has been clarified to 2 sites.

Description of visual differences in distributions across roughness classes has been removed and the text was modified to state Kolmogorov-Smirnov tests of the individual distributions for the respective variables showed significant differences across the roughness classes (p-value < 0.001).

Representative roughness values were cited from Brutsaert (1982) which indicate 0.4

m for trees up to 10 m tall. In our case we state the narrow rings of tall shrubs and trees varied between 3 m and 10 m. They also have a limited spatial footprint compared to more uniform and extensive cover for which a larger roughness length may apply.

Title for section 3.4, now labelled 3.5 due to reorganisation of the manuscript, has been changed to 'Exploratory analysis of surface variables and evaporation estimates'.

———————————

---

## Author Response (AR1)

We are grateful for the comments from Reviewer 1. The following responses address key edits which have been made to the manuscript for the comments given, including: restructuring the manuscript, discussion on potential implications and limitations, further details on energy flux and EC data, and modelling uncertainty.

Regarding comments related to the EC data:

More information has been given regarding the EC measurements in the field observation section 3.2. Such as fluxes are reported as 15 min averages, data filtering was used but there were no bad data or spikes observed on the day for which measurements were used. As such, all flux samples were accounted for and no gap filling was required. Use of the planar-fit axis rotation method to correct the latent heat flux (and sensible heat flux) measurements was also previously referred to in section 3.2. Given the corrections applied and that no missing data or spikes were observed, the fluxes were considered to be of good quality for the case study. Unfortunately, post-processing of the raw data is not possible at this stage to generate quality flags according to the methods of Mauder and Foken 2004.

Partitioning of the energy fluxes is now discussed in section 4.6 where a comparison is made between the estimated evaporation rate from a linear transect upwind of the EC station and the measured EC flux. The linear transect has been overlayed on the map in Figure 9. The text referring to the flux contributions from the upwind fetch has been clarified. Specifically it has been noted that 80% of the upwind contribution is expected to come from within 100 m of the EC station, based on a cumulative flux calculation with the model of Scheupp et al, 1990. This is along a similar linear transect used for averaging the G-D model estimates upwind of the EC station which has now been clarified in the text.

The mean daily flux values in W/m$^2$ provide the basis for testing energy balance closure as  (LE + HE) / (Q* – Qg) = (63 + 55) / (144 – 2) which gives a closure of 83%. The Bowen Ratio of 0.87 is reasonable in this semi-arid landscape due to drying of the upwind grass surface and reduced photosynthesis of the grasses as anthesis is typically in mid-late June. Uncertainty in over estimating evaporation due to neglect of the ground heat flux and possibly under measured fluxes is now referred to in section 4.6.

Regarding comments related to modelling assumptions and uncertainty:

Uncertainty associated with the modelling assumptions has been partly addressed by moving the surface reference parameter section (now 3.3) into the methods and modifying the text to provide clarity regarding the assumptions applied. A new section (4.10) was added to further discuss the general uncertainty of the methods applied. This includes discussion related to regions where the model may perform more poorly due to neglect of the ground heat flux or where wind speeds may vary due to the changes in the roughness length of the surface elements.

The relatively small magnitude of the ground heat fluxes under continuous tall grass surfaces at the two measurement sites has now been discussed in section 3.3 and is referred to again in section 4.10. A focus of this study was on the potential to scale measured values of driving energy factors across the larger field based on observable surface properties that relate directly to net radiation. Scaling ground heat flux is not possible with remote sensing methods without further detailed information on the soil moisture and density in each pixel, and application of a numerical model. Estimates could be produced at every pixel through modelling but the uncertainty may be larger than the error of the estimates reported for the net radiation estimates. It is also not uncommon to neglect the ground heat flux but implications of doing so are discussed in section 4.10.

To the best of our knowledge there were no C4 plants at the study location – all grasses/crops were cool season C3 types. Uncertainty related to prior developments of the vapour transfer function for different surface types (including C3 plants) and possible requirement of new equations for C4 plants is now noted in section 4.10.

Regarding comments related to the wind speed and turbulent energy applicable to the estimating the drying power of the air, EA:

The discussion regarding development of the surface roughness length map in section 3.4.5 has been improved for clarity and to provide evidence of considerations for potential impacts of changes in roughness and wind speed. As was stated in section 3.4.5 representative roughness lengths were selected based on reported values from Brutsaert (1982) for similar types of surfaces elements and heights for vegetation ranging between 3 – 10 m.

Regarding restructuring and combining images:

The resulting maps of albedo and Ts and the validation discussion (section 4. 1 and 4.2) are introduced earlier in the result and discussion section. Implications of relative variations in albedo and TS for estimating the net radiation and final estimates of E are discussed briefly in each of those sections.

A major restructuring was done to combine several images which reduced the number of figures from 17 to 12. The manuscript has been modified in section 4.7 and 4.7.1 and graphics have been combined so as to be more appropriate to discuss variations in the underlying distributions of driving factors and how they relate to the net radiation and impact final estimates of E. For example the underlying variability in albedo and TS appears to be much larger than the resulting variability of net radiation and E, due to the interaction of the relative evaporation, G, term.

Previous figures for the frequency distributions of evaporation and relative contributions have been combined in Fig. 10. Previous figures 12 – 14 have been combined into Fig 11 and previous figures 15 – 17 have been combined into Fig 12.

Previous figures 2 and 3 have been swapped as the discussion for figure 2 related to albedo point sampling and conversion to broad-band was moved into the methods section.

Regarding comments on covariance:

The text has been modified to clarify the impact of the relationship between relative evaporation G and net radiation. G is a non-linear function of the relative drying power D which is a function of the drying power of the air, EA and also the available energy. EA is dependent on the surface roughness length, wind speed and water vapour deficit. The non-linear inverse relationship between G and net radiation is more clearly shown and discussed via figure 12. The impact of this relationship for this study generally results in higher values of G associated with lower values of net radiation and much lower values of G associated with higher values of net radiation. The resulting interaction produced a very small covariance.

The covariance was found to be even smaller when computed within each roughness length class.

Albedo estimates which were based on the DN index showed a correlation of 0.67 with TS, which is reasonably high. However, the computed covariance was only 0.06, suggesting that that a high correlation between these variables may not be an issue.

Regarding technical corrections:

Idrisi software used to segment the data to create the roughness heights classes is now referenced, and the company is now referenced for post-processing done with Matlab software.

As indicated above several figures have been combined and discussed more appropriately.

The boxplots have been modified to remove the redundant information and the large number of data points have been overlayed and plotted with jitter.

Removed the coefficient of variation for the aerodynamic component as it appears bimodal.

Vegetation types have been more clearly defined – also figure 1 was replaced with a photo of the study region which provides more context for reference.

Tree rings clarified to narrow rings of trees.

The manuscript has been cleaned up to address typos.

The number of validation sites for net radiation has been clarified to 2 sites.

Description of visual differences in distributions across roughness classes has been removed and the text was modified to state Kolmogorov-Smirnov tests of the individual distributions for the respective variables showed significant differences across the roughness classes (p-value < 0.001).

Representative roughness length values were cited from Brutsaert (1982) which indicate 0.4 m for trees up to 10 m tall. In our case we state the narrow rings of tall shrubs and trees varied between 3 m and 10 m. They also have a limited spatial footprint compared to more uniform and extensive cover for which even larger roughness lengths may apply.

Title for section 3.4, now labelled 3.5 due to reorganisation of the manuscript, has been changed to 'Exploratory analysis of surface variables and evaporation estimates'.

We are grateful for the comments from Reviewer 2. The following responses address the comments given regarding: further details on objectives, manuscript organisation, and suggested changes in methods, results and discussion.

The study objectives and potential for advancing understanding for examining improved methods of upscaling evaporation estimates have now been clarified in the introduction / background section. The 'ratiometric' indexing methods represent a novel way of scaling single point measurements across large fields for evaporation modelling.

Portions of the text have been reorganised and edited based on the comments regarding clarity and organisation. Specifically, the last sentence of section 1 has been better integrated earlier in the introduction/background. Regarding the mix of methods and results, text related to the methods has now been worked into the methods section.

Regarding comments related to the Methods:

Repetition of the normalised index equations, i.e. equations 7-9 are referred to later in section 4.5 and Equation 10 shows the general integration for calculating net radiation – it may be preferable to leave these as is for clarity.

The reference parameters section and relevant parameters have been moved into the methods section for clarity.

A discussion of the Eddy covariance measurements and corrections has been clarified in the field observation section 3.2 and further in a new section (4.10) in relation to the modelling uncertainty.

The confusion regarding the 2006 EC study has been addressed in the text – the data collected was referenced by the Armstrong et al., 2008 study cited in the manuscript, but the year 2006 refers to when the data was collected, which has been clarified in section 3.1. The relevant text related to the methods was moved to section 3.3.

Several colour schemes were tried for the map figures but the rendered images simply did not display as well as the grey scale images used. However, Figure 1 has been replaced with an actual RGB photo of the region taken during the study flight on August 5 2007. The albedo sampling points from Figure 1 was moved to figure 2.

The replacement of Figure 1 with a colour photo provides clearer context so specific references to ponds have been removed as they can be seen clearly in the RGB photo.

The text referring to upwind fetch has been clarified to indicate that 80% of the upwind contribution comes from 100 m upwind of the EC station, based on the cumulative flux calculation with the footprint model of Scheupp et al, 1990. This is along a similar linear transect used for averaging the G-D model estimates upwind of the EC station which has now been added to Figure 9.

Regarding comments related to the Results and Discussion:

The text which describes the corrections applied to the field measurements of broadband albedo have now been moved into the methods section 3.4.3 under the methods describing the derivation of the normalised index for albedo.

It has been more clearly stated the study was relatively cloud free – the observations show just two 15 min periods later in the day when clouds passed over.

With respect to the Figure 7 relationships - the regression equation was removed from the figure. The r-square was left as it simply reflects the validity of assuming net radiation at midday can be used for temporal scaling of mean daily net radiation which appears to be more stable under clear sky conditions.

The basis for comparing the estimates and measured evaporation in section 4.7, now section 4.6, has been stated in terms of % error in the overestimate to be more relevant.

The purpose of section 4.8 and 4.8.1 (now 4.7. and 4.7.1) has been addressed to improve clarity and several figures have been combined into a single figure for more clarity and relevance to the discussion.

In the interest for avoiding confusion on the general notation of the evaporation equation (Eq. 1) and rearrangement to obtain contributions from the individual components (Eq. 14) it may be preferred to keep the two equations separate, but if crucial this can be changed.

Typographical errors have been re-checked and edited where found. It is also noted that British English spellings are also being applied throughout.

The number of figures has been reduced from 17 to 12 by combining figures 10 – 11 (now fig 10), 12 – 14 (now fig 11), and 15 – 17 (now fig 12).

List of Revisions (collated from responses to reviewer comments)

- *Please note key revisions have been documented in the response to reviewers sections but have been collated here reference purposes.*

- General edits
  - Text revisions for clarification and address typos etc.
  - Removed term 'simple' ratios for more meaningful 'ratiometric' term to describe indexing method
  - Tree rings changed to narrow rings of trees
  - Number of validation sites clarified as 2
  - Section numbers revised

- Major edits
  - Reorganisation of text related to methods found in other sections
  - Changes to headings of some sections
  - Edits to figures, combining several figures, and replacement of figure 1

- Abstract
  - New work campus location updated

- 1 Background and Introduction
  - Objectives and potential for advancing understanding for examining improved methods of upscaling evaporation estimates has been clarified
  - Text related to methods moved to methods section

- 2 Study Area
  - Figure 1 replaced with a colour photo of the study region
  - Grasses and crops clarified as C3 types

- 3 Data and Methods
  - Section 3.1
    - Some general text edits for clarity
    - Previous text related to methods from background/introduction now been worked into methods section
    - Reference parameters section and relevant parameters have been moved into the methods section
    - Collection of 2006 EC data as cited in Armstrong et al 2008 paper clarified
  - Section 3.2
    - Further information and clarification on Eddy Covariance measurements and corrections
  - Section 3.3
    - Major revision to organise reference parameters
    - Clarified relevance of 2006 EC data (as cited in Armstrong et al 2008) for examining sensitivity of model parameterisation related to current paper
  - Section 3.4
    - Changes made to sub section titles and some general edits

- Text describing the corrections applied to the field measurements of broadband albedo moved to this section (3.4.3)
- Change to figure order and content (3.4.3)
- Minor edits in 3.4.4
- Discussion regarding development of the surface roughness length map in section 3.4.5 has been improved for clarity and to provide evidence of considerations for potential impacts of changes in roughness and wind speed
  - Section 3.5
    - Title revised for clarity
    - Stated here that jitter now used for point overlay on boxplots

- 4 Results and Discussion
  - Moved surface reference parameters section (previously 5.1) into methods
  - Description of approach to albedo validation moved from section 4.1 to methods
  - Expected response of evaporation estimates to changes in albedo described (4.1)
  - Expected response of evaporation estimates to changes in TS described (4.2)
  - Clarification added in surface roughness length map section (4.3)
  - Clarification added in section 4.4 on sensitivity analysis of evaporation ratio
  - Title modified for section 4.5 and some general edits for clarity, manufacturer added for NR Lite radiometer
  - More description added in section 4.6
    - Referring to upwind fetch indicating that 80% of the upwind contribution comes from 100 m upwind of the EC station, based on the cumulative flux calculation with the footprint model of Scheupp et al, 1990. Linear transect for this added to figure 9.
    - Description on uncertainty added for EC fluxes
    - % error in the evaporation overestimate now stated to be more relevant
  - Major revision in section 4.7 to better describe purpose and clarify results for the distributions
    - Several figures combined to reduce overall number and to improve text
  - Major revision in section 4.7.1 for same purpose as section 4.7 but for roughness length classes
  - Extensive revisions in section 4.8 for including combining of figures to improve discussion of results
  - Major revision resulted in new section 4.10 to discuss uncertainty in methods

- 5 Summary and conclusions
  - Edits made for clarification and to report on % overestimate based on edits in results

[revised manuscript text omitted]
}$ | 0.134 | 0.113 | 5.28 | 0.063 | 18.70 | 0.77 | 1.03 | 1.71 | 2.73 | 2.77 | -0.03 |

---

## Author Response (AR2)

Comments regarding minor revisions from the Editor Decision on Sept 24 2019 are addressed below.

Editor Decision: Publish subject to minor revisions (further review by editor) (24 Sep 2019)

Reviewer comments and author responses:

1. I think it would be useful to show the relationship between albedo and surface temperature. A >15degC variation in surface temperature and a constant ground heat flux makes me suspicious of their assumption that Rn variability is directly proportional to L_up. The error from this assumption is probably much less than the total error they report, so not worth making major revisions about, but it is likely to be spatially significant. It would have been nice if they included this in their new discussion about the uncertainties in the modelling since it could be a useful area of future analysis.

    - The relationships between albedo ~ Ts and Rn (W/m2) ~ Ts obtained from the gridded data is shown below. The albedo ~ Ts graphic does not seem to contain any new information that might significantly enhance the paper by including it in the discussion. The variability of the 15 degC temp range is spread across a wide albedo range (including noise). A regression line for all data has been included for reference and *the relationship for the largest density of data points appears to be linear except at the lower end of the Ts range which contains the largest noise*, but this information does not seem to be noteworthy.

[Figure]

[Figure]

2. A few sections could still be cleared up.
    a. The example in section 4.4. seems arbitrary. Some variables are discussed the full range and others they pick random subsets.
        - Specific examples (i.e. 'For example, a relative increase…') have been highlighted to illustrate the impacts of required changes in key driving variables that resulted in notable changes in the evaporation ratio.
        - Text has been added to clarify this at Ln 1-2 Pg. 12.

    b. In section 4.5 (pg 12 ln 21) the authors discuss a result never presented to the reader (when Rn > 400 Wm-2).
        - This 'non-result' has now been deleted

    c. The lines on Figure 7 look like they are forced through the origin, but they don't describe what they are in the caption.
        - Yes, lines were forced through the origin to illustrate the relationships still hold with Y-intercept set to 0, and thence the proportionality
        - Caption modified to indicate what these lines represent
        - Further, text has been modified to clarify this on Pg 12 Ln 20-22

    d. Figure 12's x-scales are stretched out for the net radiation and evaporation subplots. I think they are meant to be comparable, but the authors never compare them directly.
        - Text was modified (Ln 5-6 pg. 15) to compare these subplots directly… "In this case, a large reduction in the variability of the evaporation estimates is clearly evident when compared to the same plot for net radiation."

    e. There are a number of typos still,
        i. an extra 'and' (Ln 29 pg. 11),
            - Original text at the line referenced "…compared to more broad regions of the fallow and cropped areas and grasses"
            - the first 'and' now changed to a comma (i.e. fallow, cropped areas and grasses)

        ii. Figure 3 is referenced before Figure 2,
            - Fixed - the reference to 'see Fig. 3' (Ln 16 Pg. 5) in the observations section has been deleted as it is referenced later on page 8 after the reference to Figure 2

        iii. there is a missing ')' (Ln 31 pg. 7),
            - Added the missing ')'

        iv. and figures are referenced as both 'Figure X' and 'Fig. X' without any consistency.

- Please note that referencing of the figures has been done as per the HESS guidelines to authors for manuscript preparation which is to use 'Figure X' at the beginning of a sentence and using 'Fig. X' in running text

- See guidelines https://www.hydrology-and-earth-system-sciences.net/for_authors/manuscript_preparation.html:
  - "The abbreviation "Fig." should be used when it appears in running text and should be followed by a number unless it comes at the beginning of a sentence, e.g.: "The results are depicted in Fig. 5. Figure 9 reveals that...".""

- The consistency of all figure references using the HESS guidelines has been verified

Some additional edits made:

- Edit to author affiliation Pg 1 Ln 4-6

- 4 Additional typos found and cleaned up:
  - Pg 9 Ln 19 'aeordynamic roughnesss'
  - Pg 10 Ln 16 'box whiskers. .'
  - Changed 'X%' to 'X %' in two places on Pg 13 (Ln 19 & 25)

List of Revisions (collated from addressing the minor revisions)

- Suggested edits considered but no further changes made
  - Albedo ~ Ts relationship
    - A plot of this relationship was generated from the gridded data and is shown in the response document for reference but does not appear to provide any further useful information – so no further changes have been made to the discussion section.
  - The consistency of all figure references using the HESS guidelines has been verified (see response notes in previous section)

- Edit made to Author Affiliation section on Pg 1 (Ln 4-6)
  - Affiliation updated to reflect where all work for study and manuscript preparation was done, and new university employer for past 4.5 years under which major revisions were completed.

- Minor edits as suggested by reviewer
  - Text modified in section 4.4 to specify the examples selected are for illustrative purpose (Ln 1-2 Pg. 12)
  - 'Non-result' deleted from section 4.5 (pg 12 ln 21)
  - Figure 7 caption modified to describe lines. Text modified on Pg 12 (Ln 20-22) for further clarification.
  - Regarding Figure 12, the text was modified on Pg 15 (Ln 5-6) to compare the net radiation and evaporation subplots directly.

- Typos cleaned up
  - extra 'and'  Pg. 11 (Ln 29) changed to comma
  - Fixed reference order for Figure 3 and 2 on Pg. 5 (Ln 16)
  - Added the missing ')'  Pg 7 (Ln 31)
  - Pg 9 (Ln 19) 'aeordynamic roughnesss' corrected
  - Pg 10 (Ln 16) 'box whiskers. .'
  - Pg 13 (Ln 19 & 25) changed 'X%' to 'X %'

[revised manuscript text omitted]

Surface Variations in the aerodynamic roughnessroughness length, $z_o$ and wind speed is aare important critical component factors considered in the turbulent transfer function used for for calculating the aerodynamic terms calculationsof the G-D model (Eq. 2 and 3). Previous developments (e.g. P-omeroy et al., 1997) have considered boundary layer (friction velocity) and surface parameters (roughness length) estimated from EC measurements and wind profiles over cold region, boreal forests and Canadian Pprairie region land covers such as bare soil, and C3 type crop and grasses.

Due to the general surface complexity at the study area For our purposesroughness classes for $z_o$ were needed to adjust the vapour transfer function to reflect potential increases or decreases in turbulent exchanges as result of variations in surface properties and local roughness. For example, lower values of $z_o$ associated with fallowed areas and crops would imply increased wind speeds near the ground and a reduction in turbulent energy. In contrast, larger $z_o$ values associated with taller dense grasses, shrubs and trees would effectively reduce wind speeds and increase turbulent exchanges.

, $z_o$ was needed for calculting the "drying power" term in the G-D model. In this case, For our the purposes here, a roughness length classification map for $z_o$ values was derived from the 8-bit grayscale image used for estimating surface albedos. This was acheivedachieved based on knowledge of the various land covers heights at the site and and segmentation analysis using the IDRISI Kilimanjaro surface analysis tool (Clark Labs, Clark University, Worcester, Massachusetts).

Greyscale *DNs* were initially classified into 13 zones of similarity and a segmentation analysis was applied. The method computes a standard deviation for each pixel using a 3x3 moving window filter; the standard deviation and associated *DN* for each pixel is then sorted (low to high) and a bin range assigned; a class width tolerance was set for pixels having similar standard deviations and all values within a specified range were assigned to the same class. Where pixel values were outside the range, but class boundaries overlapedoverlapped, a mid-point was determined and a new class is created.

The initial 13 classes were manually reclassified into three general classes (fallowed/cropped, grassed, and tree srings) based on a the extents of the visual comparison of dominant land cover types observed in the original and classified images. Representative Characteristic roughness heightslengths, $z_o$ were then assigned toselected for each class based onbased on standard values reported for similar surface types conditions (Brutsaert, 1982).

In this case, a value of ;-0.05 m was used for the fallowed/cropped class and 0.10 m for the taller more dense grass class0.10 m for grassed areas. Narrow rings of s

The hrubs and trees around wetlands tree rings were also dense and much relatively 
[revised manuscript text omitted]

**4.84.7 Distributions of evaporation and driving surface variables**

The following sections briefly discuss the statistical distributions of the driving variables obtained from the images used for estimating evaporation and their impacts on the resulting evaporation estimates. A key advantage of methods appliedThe benefit here is the statistical distributions of evaporation and key driving factors are considered to be physically meaningfulmeaningful.

Figure 10 indicates shows the frequency distributions of evaporation estimates and relative contributions for the energy balance and aerodynamic componentsfrom images taken on Aug 5. G-D model were evaporation estimates appear normally distributed with a and . Tthe G-D model calculated an averagemean 
[revised manuscript text omitted]
}$ | 0.134 | 0.113 | 5.28 | 0.063 | 18.70 | 0.77 | 1.03 | 1.71 | 2.73 | 2.77 | -0.03 |